# On the Computational Complexity of Private High-dimensional Model Selection

**Saptarshi Roy**    **Zehua Wang**    **Ambuj Tewari**
Department of Statistics
University of Michigan, Ann Arbor
{roysapta, wangzeh, tewaria}@umich.edu

## Abstract

We consider the problem of model selection in a high-dimensional sparse linear regression model under privacy constraints. We propose a differentially private (DP) best subset selection method with strong statistical utility properties by adopting the well-known exponential mechanism for selecting the best model. To achieve computational expediency, we propose an efficient Metropolis-Hastings algorithm and under certain regularity conditions, we establish that it enjoys polynomial mixing time to its stationary distribution. As a result, we also establish both approximate differential privacy and statistical utility for the estimates of the mixed Metropolis-Hastings chain. Finally, we perform some illustrative experiments on simulated data showing that our algorithm can quickly identify active features under reasonable privacy budget constraints.

## 1 Introduction

In this paper, we consider the problem of *private model selection* in high-dimensional sparse regression which has been one of the central topics in statistical research over the past decade. Consider $n$ observations $\{(\mathbf{x}_i, y_i)\}_{i=1}^n \subseteq \mathcal{X} \times \mathcal{Y}$ following the linear model:

$$y_i = \mathbf{x}_i^\top \boldsymbol{\beta} + w_i, \quad i \in \{1, \ldots, n\}, \tag{1}$$

where $\{\mathbf{x}_i\}_{i \in [n]}$ are *fixed* $p$-dimensional feature vectors, $\{w_i\}_{i \in [n]}$ are i.i.d. *mean-zero* $\sigma$-sub-Gaussian noise, i.e., $\mathbb{E} \exp(\lambda w_i) \leq \exp(\lambda^2 \sigma^2/2)$ for all $\lambda \in \mathbb{R}$ and $i \in [n]$, and the signal vector $\boldsymbol{\beta} \in \mathbb{R}^p$ is unknown but is assumed to have a sparse support. In matrix notation, the observations can be represented as

$$\mathbf{y} = \mathbf{X}\boldsymbol{\beta} + \mathbf{w},$$

where $\mathbf{y} = (y_1, \ldots, y_n)^\top$, $\mathbf{X} = (\mathbf{x}_1, \ldots, \mathbf{x}_n)^\top$, and $\mathbf{w} = (w_1, \ldots, w_n)^\top$. We consider the standard *high-dimensional sparse* setup where $n < p$, and possibly $n \ll p$, and the vector $\boldsymbol{\beta}$ is sparse in the sense that $\|\boldsymbol{\beta}\|_0 := \sum_{j=1}^p \mathbb{1}(\beta_j \neq 0) = s$, which is much smaller than $p$. The main goal of variable selection is to identify the active set $\gamma^* := \{j : \beta_j \neq 0\}$.

For the past two decades, there has been ample work on model selection problem in the non-private setting for $\ell_1$-penalized methods[57, 54, 46, 23], concave regularized methods [55, 53, 12, 16], $\ell_0$-penalized/constrained methods [13, 1, 37, 39] in high-dimensional setting. On the computational side, recent advancements related to mixed integer optimization (MIO) in [4, 5] and [20] have pushed the computational barrier of best subset selection (BSS) in terms of solving problems of large dimensions (large $p$), and consequently, simulation studies in [19] have revealed the improved performance of BSS over its computational surrogates like LASSO, SCAD, and MCP.

Despite these theoretical and computational advancements related to BSS, to the best of our knowledge, there is no computationally efficient private algorithmic framework for BSS for high-dimensional sparse regression setup (1). This is especially surprising as private model selection

is important in many contemporary applications involving sensitive data including genetics [21], neuroimaging [33], and computer vision [56]. One major reason for this could be the lack of DP mechanisms for MIO problems which restricts us from exploiting the MIO formulation of BSS introduced in [4]. Secondly, the apparent computational burden stemming from the requirement of exponentially large numbers of search queries in private BSS has eluded the majority of the machine learning and statistics community. In this paper, we address the latter issue by mainly focusing on the utility and computational complexity of BSS under privacy constraints. To be specific, we make the following contributions listed below:

1. We adopt the exponential mechanism [31] to design a DP BSS algorithm, and we establish its good statistical or utility guarantee under high-privacy regime whenever $\beta_{min} := \min_{j \in \gamma^*} |\beta_j| \gtrsim \sigma\{(s \log p)/n\}^{1/2}$.

2. Under the low-privacy regime, we show that accurate model recovery is possible whenever $\beta_{min} \gtrsim \sigma\{(\log p)/n\}^{1/2}$, which is the minimax optimal $\beta_{min}$ requirement for model recovery under non-private setting. Therefore, this paper points out an inflection phenomenon in the signal strength requirement for the model consistency across different privacy regimes.

3. In addition, we design an MCMC chain that converges to its stationary distribution that matches the sampling distribution in the exponential mechanism. As a consequence, the model estimator generated by the MCMC also enjoys (approximate) DP. Furthermore, under certain regularity conditions on the design, we show that the MCMC chain enjoys a polynomial mixing time in $(n, p, s)$ to the stationary distribution with good utility guarantee.

In summary, this paper proposes a DP version of BSS that generates a private model estimator of $\gamma^*$ with strong model recovery property within polynomial time in the problem parameters $n, p, s$. In the next section, we will discuss some prior related works on DP model selection and discuss some of their limitations.

## 1.1 Comparison with Prior Related Works

In the past decades, there has been a considerable amount of work studying DP sparse regression problems. However, most of these works focus either on empirical risk minimization [25, 44, 26, 48] or establishing $\ell_2$-consistency rate [47, 6] which are not directly related to the task of model selection. To the best of our knowledge, there are only three works considering the problem of variable selection in sparse regression problems under the DP framework, [27, 45], and [29]. Table 1 shows a clear comparison between those methods and our method. [27] proposed two algorithms under sparse regression setting. One of them is based on the exponential mechanism, which is known to be computationally inefficient due to exponentially large numbers of search queries. However, they do not analyze the algorithm under the model selection framework. Moreover, for the privacy analysis, they assume that the loss functions are bounded over the space of sparse vectors, which is rather restrictive in the linear regression setting. In comparison, our paper provides a solid model recovery guarantee (Theorem 3.5) for a similar exponential mechanism without using the bounded loss assumption. Furthermore, under a slightly stronger assumption, we design a computationally efficient MCMC algorithm that also enjoys desirable utility similar to the exponential mechanism (Theorem 4.3) under DP framework. The other algorithm in [27] is based on the resample-and-aggregate framework [36, 42]. Although computationally efficient, this method requires sub-optimal $\beta_{min}$ condition compared to Theorem 3.5. In [45], the authors introduced two concepts of stability for LASSO and proposed two PTR-based (propose-test-release) algorithms for variable selection. However, these methods have nontrivial probabilities of outputting the null (no result), which is undesirable in practice. Also, the support recovery probabilities for these methods do not approach 1 with a growing sample size even under the *strong irrepresentability condition* [57] on the design matrix. In [29], the authors proposed to use the Akaike information criterion or Bayesian information criterion coupled with the exponential mechanism to choose the proper model. However, the runtime of this algorithm is exponential and also requires stronger $\beta_{min}$ condition. As mentioned earlier, in this paper, we show that our proposed MCMC algorithm is both computationally efficient and produces approximate DP estimates of $\gamma^*$ with a strong utility guarantee under a better $\beta_{min}$ condition. One may also apply sparse vector techniques (SVT) to choose important features [43]. In this case, each feature can be associated with an appropriate choice of score function, and then apply SVT to choose the relevant features. However, the choice of the score function in high-dimensional sparse regression cases remains unclear, and moreover, it is also known that the exponential mechanism enjoys better accuracy compared to SVT [30] under such an offline setting.

Table 1: Comparison of DP model selection methods.

| Paper | Method | $\beta_{min}$ cond. | failure prob. $\to 0$ | runtime |
|---|---|---|---|---|
| [27] | Exp-Mech | NA | NA | exp |
| | Lasso + Samp-Agg | $\Omega(\sqrt{\frac{s\log p}{n^{1/2}}})$ | yes | poly |
| [45] | Lasso + Sub-samp. stability | $\Omega(\sqrt{\frac{s\log p}{n\varepsilon}})$ | no | poly |
| | Lasso + Pert. stability | $\Omega(\max\{\sqrt{\frac{s\log p}{n}}, \frac{s^{3/2}}{\varepsilon n}\})$ | no | poly |
| [29] | Exp-Mech | $\Omega(\sqrt{\max\{1, \frac{s}{\varepsilon}\}\frac{s\log n}{n}})$ | yes | exp |
| **This paper** | Exp-Mech | $\Omega(\sqrt{\max\{1, \frac{s}{\varepsilon}\}\frac{\log p}{n}})$ | yes | exp |
| | Approx. Exp-Mech via MCMC | $\Omega(\sqrt{\max\{1, \frac{s}{\varepsilon}\}\frac{\log p}{n}})$ | yes | poly |

## 2 Differential Privacy

Differential privacy requires the output of a randomized procedure to be robust with respect to a small perturbation in the input dataset, i.e., an attacker can hardly recover the presence or absence of a particular individual in the dataset based on the output only. It is important to note that differential privacy is a property of the randomized procedure, rather than the output obtained.

### 2.1 Preliminaries

In this section, we will formalize the notion of differential privacy. Consider a dataset $D := \{z_1, \ldots, z_n\} \in \mathcal{Z}^n$ consisting of $n$ datapoints in the sample space $\mathcal{Z}$. A *randomized* algorithm $\mathcal{A}$ maps the dataset $D$ to $\mathcal{A}(D) \in \mathcal{O}$, an output space. Thus, $\mathcal{A}(D)$ is a random variable on the output space $\mathcal{O}$.

For any two datasets $D$ and $D'$, we say they are *neighbors* if $|D\Delta D'| = 1$. We can now formally introduce the definition of differential privacy.

**Definition 2.1** (($\varepsilon, \delta$)-DP, [9])**.** *Given the privacy parameters $(\varepsilon, \delta) \in \mathbb{R}^+ \times \mathbb{R}^+$, a randomized algorithm $\mathcal{A}(\cdot)$ is said to satisfy the $(\varepsilon, \delta)$-DP property if*

$$\mathbb{P}(\mathcal{A}(D) \in \mathcal{K}) \le e^\varepsilon \mathbb{P}(\mathcal{A}(D') \in \mathcal{K}) + \delta \tag{2}$$

*for any measurable event $\mathcal{K} \in range(\mathcal{A})$ and for any pair of neighboring datasets $D$ and $D'$.*

In the above definition, the probability is only with respect to the randomness of the algorithm $\mathcal{A}(\cdot)$, and it does not impose any condition on the distribution of $D$ or $D'$. If both $\varepsilon$ and $\delta$ are small, then Definition 2.1 essentially entails that distribution of $\mathcal{A}(D)$ and $\mathcal{A}(D')$ are almost indistinguishable from each other for any choices of neighboring datasets $D$ and $D'$. This guarantees strong privacy against an attacker by masking the presence or absence of a particular individual in the dataset. As a special case, when $\delta = 0$, the notion of DP in Definition 2.1 is known as the *pure differential privacy*.

### 2.2 Privacy Mechanisms

For any DP procedure, a specific randomized procedure $\mathcal{A}$ must be designed that takes a database $D \in \mathcal{Z}^n$ as input and returns an element of the output space $\mathcal{O}$ while satisfying the condition in (2). Several approaches exist that are generic enough to be adaptable to different tasks, and which often serve as building blocks for more complex ones. A few popular examples include the Laplace mechanism [11], Gaussian mechanism [10], and Exponential mechanism [31]. We only provide the details of the last technique, since the other two techniques are out-of-scope for the methods and experiments in this paper.

**Exponential mechanism:** The exponential mechanism is designed for discrete output space, Suppose $\mathscr{S} = \{\alpha_i : i \in \mathcal{I}\}$ for some index set $\mathcal{I}$, and let $u : \mathscr{S} \times \mathcal{Z}^n \to \mathbb{R}$ be score function that measures the quality of $\alpha \in \mathscr{S}$. Denote by $\Delta u_K$ the global sensitivity of the score function $u$, i.e.

$$\Delta u_K := \max_{\alpha \in \mathscr{S}} \max_{D, D' \text{ are neighbors}} |u(\alpha, D) - u(\alpha, D')|.$$

Intuitively, sensitivity quantifies the effect of any individual in the dataset on the outcome of the analysis. The score function $u(\cdot, \cdot)$ is called *data monotone* if the addition of a data record can either increase (decrease) or remain the same with any outcome, e.g., $u(\alpha, D) \leq u(\alpha, D \cup \{z\})$. Next, we have the following result.

**Lemma 2.2** ([8, 31]). *Exponential mechanism $\mathcal{A}_E(D)$ that outputs samples from the probability distribution*

$$\mathbb{P}(\mathcal{A}_E(D) = \alpha) \propto \exp\left\{ \frac{\varepsilon u(\alpha, D)}{\Delta u} \right\} \tag{3}$$

*preserves $(2\varepsilon, 0)$-differential privacy. If $u(\cdot, \cdot)$ is data monotone, then we have $(\varepsilon, 0)$-differential privacy.*

In general, if the $\mathscr{S}$ is too large, the sampling from the distribution could be computationally inefficient. However, we show below that the special structure of the linear model (1) allows us to design an MCMC chain that can generate approximate samples *efficiently* from the distribution (3) for privately solving BSS under an appropriately chosen score function.

## 3 Best Subset Selection

We briefly review the preliminaries of BSS, one of the most classical variable selection approaches. For a given sparsity level $\widehat{s}$, BSS solves for $\widehat{\boldsymbol{\beta}}_{\text{best}}(\widehat{s}) := \arg\min_{\boldsymbol{\theta} \in \mathbb{R}^p, \|\boldsymbol{\theta}\|_0 \leq \widehat{s}} \|\mathbf{y} - \mathbf{X}\boldsymbol{\theta}\|_2^2$. For model selection purposes, we can choose the best fitting model to be $\widehat{\gamma}_{\text{best}}(\widehat{s}) := \{j : [\widehat{\boldsymbol{\beta}}_{\text{best}}(\widehat{s})]_j \neq 0\}$. For a subset $\gamma \subseteq [p]$, define the matrix $\mathbf{X}_\gamma := (\mathbf{X}_j; j \in \gamma)$. Let $\boldsymbol{\Phi}_\gamma := \mathbf{X}_\gamma(\mathbf{X}_\gamma^\top \mathbf{X}_\gamma)^{-1}\mathbf{X}_\gamma^\top$ be orthogonal projection operator onto the column space of $\mathbf{X}_\gamma$. Also, define the corresponding residual sum of squares (RSS) for model $\gamma$ as $L_\gamma(\mathbf{y}, \mathbf{X}) := \mathbf{y}^\top(\mathbb{I}_n - \boldsymbol{\Phi}_\gamma)\mathbf{y}$. With this notation, the $\widehat{\gamma}_{\text{best}}(\widehat{s})$ can be alternatively written as

$$\widehat{\gamma}_{\text{best}}(\widehat{s}) := \arg\min_{\gamma \subseteq [p]:|\gamma| \leq \widehat{s}} L_\gamma(\mathbf{y}, \mathbf{X}). \tag{4}$$

Let $\mathbf{X}_\gamma$ be the matrix comprised of only the columns of $\mathbf{X}$ with indices in $\gamma$, and $\boldsymbol{\Phi}_\gamma$ denotes the orthogonal projection matrix onto the column space of $\mathbf{X}_\gamma$. In addition, let $\widehat{\boldsymbol{\Sigma}} := n^{-1}\mathbf{X}^\top\mathbf{X}$ be the sample covariance matrix and for any two sets $\gamma_1, \gamma_2 \subset [p]$, $\widehat{\boldsymbol{\Sigma}}_{\gamma_1, \gamma_2}$ denotes the submatrix of $\boldsymbol{\Sigma}$ with row indices in $\gamma_1$ and column indices in $\gamma_2$. Finally, define the collection $\mathscr{A}_{\widehat{s}} := \{\gamma \subset [p] : \gamma \neq \gamma^*, |\gamma| = \widehat{s}\}$, and for $\gamma \in \mathscr{A}_{\widehat{s}}$ write $\Gamma(\gamma) = \widehat{\boldsymbol{\Sigma}}_{\gamma^*\backslash\gamma, \gamma^*\backslash\gamma} - \widehat{\boldsymbol{\Sigma}}_{\gamma^*\backslash\gamma, \gamma}\widehat{\boldsymbol{\Sigma}}_{\gamma, \gamma}^{-1}\widehat{\boldsymbol{\Sigma}}_{\gamma, \gamma^*\backslash\gamma}$. Then, it follows that $\boldsymbol{\beta}_{\gamma^*\backslash\gamma}^\top\Gamma(\gamma)\boldsymbol{\beta}_{\gamma^*\backslash\gamma}$ is equal to the *residualized* signal strength $n^{-1}\|(\mathbb{I}_n - \boldsymbol{\Phi}_\gamma)\mathbf{X}_{\gamma^*\backslash\gamma}\boldsymbol{\beta}_{\gamma^*\backslash\gamma}\|_2^2$. Therefore, $\boldsymbol{\beta}_{\gamma^*\backslash\gamma}^\top\Gamma(\gamma)\boldsymbol{\beta}_{\gamma^*\backslash\gamma}$ quantifies the separation between $\gamma$ and the true model $\gamma^*$. Ideally, a larger value of the quantity will help BSS to discriminate between $\gamma^*$ and any other candidate model $\gamma$. More details on this can be found in [40]. Now we are ready to introduce the identifiability margin that characterizes the *model discriminative power* of BSS.

### 3.1 Identifiability Margin

The discussion in Section 3 motivates us to define the following *identifiablity margin*:

$$\mathfrak{m}_*(\widehat{s}) := \min_{\gamma \in \mathscr{A}_{\widehat{s}}} \frac{\boldsymbol{\beta}_{\gamma^*\backslash\gamma}^\top\Gamma(\gamma)\boldsymbol{\beta}_{\gamma^*\backslash\gamma}}{|\gamma \backslash \gamma^*|}. \tag{5}$$

As mentioned earlier, the quantity $\mathfrak{m}_*(\widehat{s})$ captures the model discriminative power of BSS. To add more perspective, note that if the features are highly correlated among themselves then it is expected that $\mathfrak{m}_*(\widehat{s})$ is very close to 0. Hence, any candidate model $\gamma$ is practically indistinguishable from the true model $\gamma^*$ which in turn makes the problem of exact model recovery harder. On the contrary, if the features are uncorrelated then $\mathfrak{m}_*(\widehat{s})$ becomes bounded away from 0 making the true model $\gamma^*$ easily recoverable. For example, [17] showed that under the knowledge of true sparsity, i.e., when $\widehat{s} = s$, the condition

$$\mathfrak{m}_*(s) \gtrsim \sigma^2 \frac{\log p}{n}, \tag{6}$$

is sufficient for BSS to achieve model consistency. One can also view $\mathfrak{m}_*(s)$ as a quantifier of the coupled effect of model correlation and signal strength. If we define the minimum and maximum

eigenvalues over all models to be $\lambda_* = \min_{\gamma \in \mathscr{A}_s} \lambda_{min}(\Gamma(\gamma))$ and $\lambda^* = \max_{\gamma \in \mathscr{A}_s} \lambda_{max}(\Gamma(\gamma))$ respectively, then it follows that

$$\lambda_* \beta_{min}^2 \leq \mathfrak{m}_*(s) \leq \lambda^* \beta_{min}^2.$$

Therefore, it suffices to have $\beta_{min} \gtrsim \sigma\{(\log p)/(n\lambda_*)\}^{1/2}$ in order to satisfy condition (6). In this case, $\lambda_*$ captures the degree of model correlation, and $\beta_{min}$ is the minimum signal strength. Similar to our previous discussion, if there is high collinearity in the model, $\lambda_*$ will be typically small, and BSS needs a large value of $\beta_{min}$ to identify the true model $\gamma^*$. On the other hand, if $\beta_{min}$ is too small for a given level of model correlation, i,e, if $\beta_{min} \ll \sigma\{(\log p)/(n\lambda^*)\}^{1/2}$, then also BSS fails to achieve model consistency as it is hard to identify active features under the presence of weak signals [17, Theorem 2.1]. As we will see in the next section, the DP BSS algorithm also requires a margin condition similar to (6) to ensure model recovery, and this is indeed an indispensable condition as it is needed even in non-private case.

## 3.2 Differentially Private BSS and Utility Analysis

In order to privatize the optimization problem in (4), we will adopt the exponential mechanism discussed in Section 2.2. In particular, for a tuning parameter $K > 0$, we consider the score function

$$u_K(\gamma; \mathbf{X}, \mathbf{y}) := - \min_{\boldsymbol{\theta} \in \mathbb{R}^s : \|\boldsymbol{\theta}\|_1 \leq K} \|\mathbf{y} - \mathbf{X}_\gamma \boldsymbol{\theta}\|_2^2 \,,$$

and for a given privacy budget $\varepsilon > 0$, we sample $\gamma \in \mathscr{A}_{\widehat{s}}$ from the distribution

$$\pi(\gamma) \propto \exp\left\{ \frac{\varepsilon u_K(\gamma; \mathbf{X}, \mathbf{y})}{\Delta u_K} \right\} \mathbb{1}(\gamma \in \mathscr{A}_{\widehat{s}} \cup \{\gamma^*\}). \tag{7}$$

As we are concerned with the exact recovery $\gamma^*$, from here on we assume $\widehat{s} = s$. The above algorithm is essentially the same as Algorithm 4 in [27]; however, they do not introduce the extra $\ell_1$-constraint on the parameter space. Instead, their algorithm needs the loss-term $(y - \mathbf{x}_\gamma^\top \boldsymbol{\theta})^2$ to be bounded by a constant for every possible choice of $\mathbf{x}, y, \gamma$ and $\boldsymbol{\theta}$. This assumption is not true in general for the squared error loss, and to remedy this issue, we introduce the extra $\ell_1$-constraint in the score function. This is a common strategy that is used to guarantee worst-case sensitivity bound and similar methods also have been adopted in [29, 6] to construct private estimators. Next, we present the following lemma that shows the data-monotonicity of the proposed score function.

**Lemma 3.1.** *The score function $u_K(\gamma; \cdot)$ in (7) is data monotone.*

Therefore, Lemma 2.2 automatically guarantees that the above procedure is $(\varepsilon, 0)$-DP. However, in practice, we need an explicit form for $\Delta u_K$ to carry out the sampling method, and it is also needed to analyze the utility guarantee of the exponential mechanism. To provide a concrete upper bound on the global sensitivity of $u_K(\cdot; \cdot)$, we make the following boundedness assumption on the database:

**Assumption 3.2.** *There exists positive constants $r, x_{\mathsf{max}}$ such that $\sup_{y \in \mathcal{Y}} |y| \leq r, \sup_{\mathbf{x} \in \mathcal{X}} \|\mathbf{x}\|_\infty \leq x_{\mathsf{max}}$.*

Under this assumption, the following lemma provides an upper bound on the global sensitivity of the score function along with the DP guarantee.

**Lemma 3.3** (Sensitivity bound and DP). *Under Assumption 3.2, the global sensitivity $\Delta u_K$ is bounded by $\Delta_K := (r + x_{\mathsf{max}} K)^2$. Therefore, the exponential mechanism (7) with $\Delta u_K$ replaced by $\Delta_K$ satisfies $(\varepsilon, 0)$-DP.*

The above lemma provides an upper bound on the global sensitivity of the score function rather than finding the exact value of it. However, to guarantee $(\varepsilon, 0)$-DP property of exponential mechanism, it suffices to use the upper bound of $\Delta u_K$ in (7). Now we will shift towards the utility analysis of the proposed exponential mechanism. First, we require some technical assumptions.

**Assumption 3.4.** *We assume the following hold:*

(a) *There exists positive constants $b_{\mathsf{max}}$ such that $\|\boldsymbol{\beta}\|_1 \leq b_{\mathsf{max}}$.*

(b) *There exists positive constants $\kappa_-, \kappa_+$ such that*

$$\kappa_- \leq \lambda_{\min}\left(\mathbf{X}_\gamma^\top \mathbf{X}_\gamma / n\right) \leq \lambda_{\max}\left(\mathbf{X}_\gamma^\top \mathbf{X}_\gamma / n\right) \leq \kappa_+, \tag{8}$$

*for all $\gamma \in \mathscr{A}_s \cup \{\gamma^*\}$.*

*(c) The true sparsity level $s$ follows the inequality $s \leq n/(\log p)$.*

Assumption 3.4(a) tells that the true parameter $\boldsymbol{\beta}$ lies inside a $\ell_1$-ball. Similar boundedness assumptions are fairly standard in privacy literature [49, 29, 6]. Assumption 3.4(b) is a well-known assumption in the high-dimensional literature [54, 22, 32] which is known as the Sparse Riesz Condition (SRC). Finally, Assumption 3.4(c) essentially assumes that the $s = o(n)$, i.e., sparsity grows with a sufficiently small rate compared to the sample size $n$.

**Theorem 3.5** (Utility gurantee). *Let the conditions in Assumption 3.2 and Assumption 3.4 hold. Set $K \geq \{(\kappa_+/\kappa_-)b_{\max} + (8x_{\max}/\kappa_-)\sigma\}\sqrt{s}$. Then, under the data generative model* (1)*, there exist universal positive constants $c_1, C_1$ such that whenever*

$$\mathfrak{m}_*(s) \geq C_1 \sigma^2 \max\left\{1, \frac{\Delta_K}{\varepsilon\sigma^2}\right\} \frac{\log p}{n}, \tag{9}$$

*with probability at least $1 - c_1 p^{-2}$ we have $\pi(\gamma^*) \geq 1 - p^{-2}$.*

**Cost of privacy.** Theorem 3.5 essentially says that whenever the identifiability margin is large enough, the exponential mechanism outputs the true model $\gamma^*$ with high probability. Note that $\Delta_K/\sigma^2 = \Omega(s)$. In the low privacy regime, i.e., for $\varepsilon > \Delta_K/\sigma^2$ we only require $\mathfrak{m}_*(s) \gtrsim \sigma^2(\log p)/n$ to achieve model consistency and this matches with the optimal rate for model consistency of non-private BSS. Note that, the margin condition does not depend at all on $\varepsilon$ in this regime. In contrast, in a high privacy regime, i.e., for $\varepsilon < \Delta_K/\sigma^2$, Condition (9) essentially demands $\mathfrak{m}_*(s) \gtrsim \sigma^2(s\log p)/(n\varepsilon)$ to achieve model consistency. Thus, in a high privacy regime, we pay an extra factor of $(s/\varepsilon)$ in the margin requirement.

**Remark 3.6.** *The failure probability in Theorem 3.5 can be improved to $O(p^{-M})$ for any arbitrary integer $M > 2$. However, we have to pay a cost in the universal constant $C_1$ in terms of a multiplicative constant larger than 1.*

**Remark 3.7.** *Under Assumption 3.4(b), it follows that $\lambda_* \geq \kappa_-$. Therefore, it suffices to have $\min_{j \in \gamma^*} \beta_j^2 \geq \left(\frac{C_1\sigma^2}{\kappa_-}\right)\max\left\{1, \Delta_K/(\varepsilon\sigma^2)\right\}\frac{\log p}{n}$ in order to hold condition (9). Therefore, in high-privacy regime, our method requires $\min_{j \in \gamma^*} |\beta_j| \gtrsim \sigma\{(s\log p)/(n\varepsilon\kappa_-)\}^{1/2}$. In contrast, under the low-privacy regime, we retrieve the optimal requirement $\min_{j \in \gamma^*} |\beta_j| \gtrsim \sigma\{(\log p)/(n\kappa_-)\}^{1/2}$.*

## 4 Efficient Sampling through MCMC

In this section, we will propose an efficient sampling method to generate approximate samples from the distribution (7). One of the challenges of sampling methods in high-dimension is their high computational complexity. For example, the distribution in (7) places mass on all $\binom{p}{s}$ subsets of $[p]$, and it is practically infeasible to sample $\gamma$ from the distribution as we have to essentially explore over an exponentially large space. This motivates us to resort to sampling techniques based on MCMC, through which we aim to obtain approximate samples from the distribution in (7). Past works on MCMC algorithms for Bayesian variable selection can be divided into two main classes – Gibbs sampler [15, 24, 34] and Metropolis-Hastings [18, 28]. In this paper, we focus on a particular form of Metropolis-Hastings updates.

In general terms, Metropolis-Hastings (MH) random walk is an iterative and local-move based method involving three steps:

1. Given the current state $\gamma$, construct a neighborhood $\mathcal{N}(\gamma)$ of proposal states.

2. Choose a new state $\gamma' \in \mathcal{N}(\gamma)$ according to some proposal distribution $\mathbf{F}(\gamma, \cdot)$ over the neighborhood $\mathcal{N}(\gamma)$.

3. Move to the new state $\gamma'$ with probability $\mathbf{R}(\gamma, \gamma')$, and stay in the original state $\gamma$ with probability $1 - \mathbf{R}(\gamma, \gamma')$, where the acceptance probability is given by

$$\mathbf{R}(\gamma, \gamma') = \min\left\{1, \frac{\pi(\gamma')\mathbf{F}(\gamma', \gamma)}{\pi(\gamma)\mathbf{F}(\gamma, \gamma')}\right\},$$

where $\pi(\cdot)$ is same as in Equation (7).

This procedure generates a Markov chain for any choice of the neighborhood structure $\mathcal{N}(\gamma)$ with the following transition probability:

$$\mathbf{P}_{\mathsf{MH}}(\gamma, \gamma') = \begin{cases} \mathbf{F}(\gamma, \gamma')\mathbf{R}(\gamma, \gamma'), & \text{if } \gamma' \in \mathcal{N}(\gamma), \\ 1 - \sum_{\gamma' \neq \gamma} \mathbf{P}_{\mathsf{MH}}(\gamma, \gamma'), & \text{if } \gamma' = \gamma, \\ 0, & \text{otherwise.} \end{cases}$$

The specific form of Metropolis-Hastings update analyzed in this paper is obtained by following the *double swap update* scheme to update $\gamma$.

**Double swap update:** Let $\gamma \in \mathscr{A}_s \cup \{\gamma^*\}$ be the initial state. Choose an index pair $(k, \ell) \in \gamma \times \gamma^c$ *uniformly* at random. Construct the new state $\gamma'$ by setting $\gamma' = \gamma \cup \{\ell\} \setminus \{k\}$.

The above scheme can be viewed as a general MH update scheme when $\mathcal{N}(\gamma)$ is the collection of all models $\gamma'$ which can be obtained by swapping two distinct coordinates of $\gamma$ and $\gamma^c$ respectively. Thus, letting $d_H(\gamma, \gamma') = |\gamma \setminus \gamma'| + |\gamma' \setminus \gamma|$ denote the Hamming distance between $\gamma$ and $\gamma'$, the neighborhood is given by $\mathcal{N}(\gamma) = \{\gamma' \mid d_H(\gamma, \gamma') = 2, \exists\ (k, \ell) \in \gamma \times \gamma^c \text{ such that } \gamma' = \gamma \cup \{\ell\} \setminus \{k\}\}$. With this definition, the transition matrix of the previously described Metropolis-Hastings scheme can be written as follows:

$$\mathbf{P}_{\mathsf{MH}}(\gamma, \gamma') = \begin{cases} \frac{1}{|\gamma||\gamma^c|} \min\{1, \frac{\pi(\gamma')}{\pi(\gamma)}\}, & \text{if } \gamma' \in \mathcal{N}(\gamma), \\ 1 - \sum_{\gamma' \neq \gamma} \mathbf{P}_{\mathsf{MH}}(\gamma, \gamma'), & \text{if } \gamma' = \gamma, \\ 0, & \text{otherwise.} \end{cases} \tag{10}$$

## 4.1 Mixing Time and Approximate DP

Let $\mathcal{C}$ be a Markov chain on the discrete space $\mathscr{S}$ with a transition probability matrix $\mathbf{P} \in \mathbb{R}^{|\mathscr{S}| \times |\mathscr{S}|}$ with stationary distribution $\nu$. Throughout our discussion, we assume that $\mathcal{C}$ is reversible, i.e., it satisfies the balanced condition $\nu(\gamma)\mathbf{P}(\gamma, \gamma') = \nu(\gamma')\mathbf{P}(\gamma', \gamma)$ for all $\gamma, \gamma' \in \mathscr{S}$. Note that the previously described transition matrix $\mathbf{P}_{\mathsf{MH}}$ in (10) satisfies the reversibility condition. It is convenient to identify a reversible chain with a weighted undirected graph $G$ on the vertex set $\mathscr{S}$, where two vertices $\gamma$ and $\gamma'$ are connected if and only if the edge weight $\mathbf{Q}(\gamma, \gamma') := \nu(\gamma)\mathbf{P}(\gamma, \gamma')$ is strictly positive. For $\gamma \in \mathscr{S}$ and any subset $S \subseteq \mathscr{S}$, we write $\mathbf{P}(\gamma, S) = \sum_{\gamma' \in S} \mathbf{P}(\gamma, \gamma')$. If $\gamma$ is the initial state of the chain, then the total variation distance to the stationary distribution after $t$ iterations is

$$\Delta_\gamma(t) = \left\| \mathbf{P}^t(\gamma, \cdot) - \nu(\cdot) \right\|_{\mathsf{TV}} := \max_{S \subset \mathscr{S}} \left| \mathbf{P}^t(\gamma, S) - \nu(S) \right|.$$

The $\eta$-mixing time is given by

$$\tau_\eta := \max_{\gamma \in \mathscr{S}} \min\{t \in \mathbb{N} \mid \Delta_\gamma(t') \leq \eta \text{ for all } t' \geq t\}, \tag{11}$$

which measures the number of iterations needed for the chain to be within distance $\eta \in (0, 1)$ of the stationary distribution.

**Privacy of MCMC estimator:** Now, we will show that once the MH chain in (10) has mixed with its stationary distribution $\pi(\cdot)$ defined in (7), the model estimators at each iteration will enjoy approximate DP. To fix the notation, let $\gamma_t$ be the $t$th iteration of the MH chain in (10). Then, we have the following useful lemma:

**Lemma 4.1.** *The model estimator $\gamma_{\tau_\eta}$ is $(\varepsilon, \delta)$-DP with $\delta = \eta(1 + e^\varepsilon)$.*

The above lemma shows that smaller $\eta$ entails a better privacy guarantee for a fixed level $\varepsilon$ as $\delta$ decreases with $\eta$. Therefore, allowing more mixing of the chain will provide better privacy protection. However, this raises a concern about how long a practitioner must wait until the chain archives $\eta$-mixing. In particular, it is important to understand how $\tau_\eta$ scales in the difficulty parameters of the problem, for example, the dimension of the parameter space and sample size. In our case, we are interested in the covariate dimension $p$, sample size $n$, sparsity $s$, and the privacy parameter $\varepsilon$. In the next section, we will show that the chain with transition matrix (10) enjoys rapid mixing, meaning that the mixing time $\tau_\eta$ grows at most at a polynomial rate in $p, s$ and the sample size $n$.

## 4.2 Rapid Mixing of MCMC

We now turn to develop sufficient conditions for MH scheme (10) to be rapidly mixing. To this end, we make a technical assumption on the design matrix. Essentially, the following assumption controls the amount of correlation between active features and spurious features.

**Assumption 4.2.** *For every $\gamma' \in \mathscr{A}_s \setminus \{\gamma^*\}$, there exists $k \notin \gamma^* \cup \gamma'$ such that*

$$\max_{j \in \gamma^* \setminus \gamma'} \frac{\left| \mathbf{X}_j^\top (\mathbb{I}_n - \mathbf{\Phi}_{\gamma'}) \mathbf{X}_k \right|}{\left\| (\mathbb{I}_n - \mathbf{\Phi}_{\gamma'}) \mathbf{X}_k \right\|_2} \leq b_{\mathsf{max}}^{-1} \sqrt{(\kappa_- C_1 \sigma^2/2) \log p},$$

*where $C_1$ is the same universal positive constant as in Theorem 3.5.*

First, note that Assumption 4.2 basically controls the length of the projection of the feature $\mathbf{X}_j$ on the unit vector $\mathbf{u}_k := (\mathbb{I}_n - \mathbf{\Phi}_{\gamma'}) \mathbf{X}_k / \left\| (\mathbb{I}_n - \mathbf{\Phi}_{\gamma'}) \mathbf{X}_k \right\|_2$. Therefore, the above inequality restricts the correlation between an active feature $\mathbf{X}_j$ and the spurious scaled feature $\mathbf{u}_k$ from being too large. To this end, we emphasize that stronger assumptions on model correlation (on top of the SRC condition) are common in literature for establishing the computational efficiency of Bayesian variable selection methods involving MH algorithm. For example, to show the computational efficiency MH algorithm under Zellner's $g$-prior, [51] assumes

$$\max_{\gamma: |\gamma| \leq s_0} \left\| (\mathbf{X}_\gamma^\top \mathbf{X}_\gamma)^{-1} \mathbf{X}_\gamma^\top \mathbf{X}_{\gamma^* \setminus \gamma} \right\|_{\mathsf{op}}^2 = O\left( \frac{n}{s \log p} \right), \tag{12}$$

where $s_0$ (larger than $s$) is a specific tuning parameter of their algorithm that controls the model size. The assumption in the above display is akin to the well-known irrepresentability condition [57] which is a very strong assumption on the design. On a high level, at any given current state $\gamma$, Assumption 4.2 or Condition (12) helps to identify a good local move towards the true model $\gamma^*$ in the MH algorithm via deletion of the least influential covariate in $\gamma$. Now, we present our main result for the mixing time of MCMC.

**Theorem 4.3** (Rapid mixing time). *Let the conditions in Assumption 3.2, Assumption 3.4 and Assumption 4.2 hold. Then, under the data generative model* (1)*, there exists a universal constant $C_1' > 0$ such that under the margin condition*

$$\mathbb{m}_*(s) \geq C_1' \sigma^2 \max\left\{ 1, \frac{\Delta_K}{(\kappa_- \wedge 1)\varepsilon \sigma^2} \right\} \frac{\log p}{n}, \tag{13}$$

*there exist universal positive constants $c_2, C_2$ such that the mixing time $\tau_\eta$ of the MCMC chain* (10) *enjoys the following with probability at least $1 - c_2 p^{-2}$:*

$$\tau_\eta \leq C_2 p s^2 \left\{ n\varepsilon \Psi^{-1} \kappa_+ b_{\mathsf{max}}^2 + \log(1/\eta) \right\}, \tag{14}$$

*where $\Psi = \left\{ r + (\kappa_+/\kappa_-) b_{\mathsf{max}} x_{\mathsf{max}} + (\sigma/\kappa_-) x_{\mathsf{max}}^2 \right\}^2$.*

The main technical innovation in the theorem is the double swap updating scheme in the MCMC that allows us to leverage the *canonical path ensemble* construction argument [41] to prove the bound (14) on the mixing time. Essentially, we show that under Assumption 4.2, there exists a canonical path in a specially weighted graph corresponding to the MCMC random walk with low path congestion. The complete proof can be found in Appendix A.5.

Regarding the statement of the theorem, note that the margin condition (13) is slightly stronger than the margin condition in Theorem 3.5. Under that condition, the above theorem shows that the $\eta$-mixing time of the MCMC algorithm designed for approximate sampling from the distribution (7) grows at a polynomial rate in $(n, p, s)$. Recall that according to the previous definition (11) of the mixing time, Theorem 4.3 characterizes the *worst-case* mixing time, meaning the number of iterations when starting from the worst possible initialization. If we start with a good initial state — for example, the true model $\gamma^*$ would be an ideal though impractical choice - then we can remove the $n$ term in the upper bound in (14). Therefore, the bound in (14) can be thought of as the worst-case number of iterations required in the burn-in period of the MCMC algorithm. Furthermore, it is important to point out that Assumption 4.2 is only needed to ensure the "quick" mixing time of the MCMC chain. It is possible to relax this assumption, however, in that case, the MCMC chain is not guaranteed to mix under polynomial time. Nonetheless, given enough iterations, the chain will indeed

converge to the distribution (7) as MH algorithm always generates an ergodic chain that eventually mixes to its stationary distribution.

It is interesting to note that Theorem 4.3 suggests that in a large $\varepsilon$ regime, the chain mixes slower compared to the small $\varepsilon$ regime. The main reason for this is that Theorem 4.3 only relies on worst-case analysis. The intuition is the following: When $\varepsilon$ is very large, then the target distribution is essentially fully concentrated on $\gamma^*$ (assuming the score for $\gamma^*$ is highest). Now, the current analysis of Theorem 4.3 does not assume any condition on the initial state of the MCMC chain. It treats the initial state $\gamma_0$ as if it is chosen in a completely random manner, i.e., it is the worst case. From this point of view, it is hard for a completely uninformative distribution to converge to a target distribution that is concentrated on a single subset (very informative), and resulting in a longer mixing time. Finally, Theorem 4.3 leads to the following corollary:

**Corollary 4.4.** *Let $\pi_t$ denote the distribution of the $t$th iterate $\gamma_t$ of the MCMC scheme* (10). *Then, under the conditions of Theorem 3.5 and Theorem 4.3, there exists a universal constant $c_3 > 0$ such that for any fixed iteration $t$ such that $t \geq C_2 p s^2 \left\{ n\varepsilon\Psi^{-1}\kappa_+ b_{\mathsf{max}}^2 + \log(1/\eta) \right\}$, we have $\pi_t(\gamma^*) \geq 1 - \eta - p^{-2}$ with probability at least $1 - c_3 p^{-2}$.*

The above corollary is useful in the sense that it provides a quantitative choice of $\eta$ that yields high utility of the estimator $\gamma_t$. For example, if we set $\eta = p^{-2}$ and $\varepsilon = O(1)$, then for any $t \gtrsim p s^2 (n\varepsilon + \log p)$ the resulting sample $\gamma_t$ will match $\gamma^*$ with probability $1 - c_3 p^{-2}$.

**Remark 4.5.** *Similar to Remark 3.6, the failure probability in Theorem 4.3 and Corollary 4.4 can be improved to $O(p^{-M})$ for arbitrary large $M > 2$, but at the cost of paying higher values for the absolute constants $C_1'$ and $C_2$.*

## 5  Numerical experiments

In this section, we will conduct some illustrative simulations. To compare the quality of the DP model estimator, we compare F-score [19] of the estimated model with that of the true model $\gamma^*$ and the BSS estimator. As the actual BSS is computationally infeasible, we use the adaptive best subset selection (ABESS) algorithm [58] as a computational surrogate to BSS. Throughout this section, we assume that the true sparsity $s$ is known, i.e., we provide the knowledge of $s$ to the algorithm. All codes are available at https://github.com/roysaptaumich/DP-BSS.

**Uniform design.**   We consider a random design matrix, formed by choosing each entry from the distribution $\mathrm{Uniform}(-1, 1)$ in i.i.d. fashion. In detail, we set $n = 900, p = 2000$, and the sparsity level $s = 4$. We generate the entries of the noise $\mathbf{w}$ independently from $\mathrm{Uniform}(-0.1, 0.1)$, and consider the linear model (1). We choose the design vector $\boldsymbol{\beta}$ with true sparsity $s = 4$ and the support set $\gamma^* = \{j : 1 \leq j \leq 4\}$. We set all the signal strength to be equal, taking the following two forms: (i) **Strong signal:** $\beta_j = 2\{(s \log p)/n\}^{1/2}$, and (ii) **Weak signal:** $\beta_j = 2\{(\log p)/n\}^{1/2}$ for all $j \in \gamma^*$.

Under these setups, we consider the privacy parameter $\varepsilon \in \{0.5, 1, 3, 5, 10\}$ which are acceptable choices of $\varepsilon$ [35]. Moreover, similar (or larger) choices of $\varepsilon$ are common in various applications including US census study [14], socio-economic study [38], and industrial applications [3, 52]. For the Metropolis-Hastings random walk, we vary $K \in \{0.5, 2, 3, 3.5\}$ and initialize 10 independent Markov chains from random initializations and record the F-score of the last iteration. We use the CVXPY package [7, 2] for solving the $\ell_1$-constrained optimization problem in the updating step of MCMC. We also track the qualities of the model through its explanatory power for convergence diagnostics. In particular, we calculate the scale factor $R_\gamma := \mathbf{y}^\top \boldsymbol{\Phi}_\gamma \mathbf{y} / \|\mathbf{y}\|_2^2$ for each model update along the random walk and compare those with $R_{\widehat{\gamma}_{\mathsf{best}}}$ to heuristically gauge the quality of mixing. More details and a set of comprehensive plots can be found in Appendix D.1 where we also discuss more about the effect of $\varepsilon$ and $K$ on the utility. For $K = 2$, Table 2 shows that F-score increases as $\varepsilon$ increases both in the cases of strong and weak signals. In fact, for $\varepsilon \geq 3$, the performance of the algorithm is on par with the non-private BSS. This is consistent with the inflection phenomenon pointed out in Theorem 3.5 and Corollary 4.4. Furthermore, as expected, we see that for a fixed $\varepsilon$, the F-score is generally higher in the strong signal case.

Table 2: Comparison of mean `F-score` across chains for $K = 2$ under independent Uniform and Gaussian design. ($*$) denotes that the chain has mixed reasonably well.

| Privacy Level | Uniform Design | | Gaussian Design | |
|:---:|:---:|:---:|:---:|:---:|
| | Strong Signal | Weak Signal | Strong Signal | Weak Signal |
| $\varepsilon = 0.5$ | **0.025** | 0.00 | 0.00 | 0.00 |
| $\varepsilon = 1$ | **0.15** | 0.05 | 0.00 | 0.00 |
| $\varepsilon = 3$ | **1.00*** | 0.15 | **0.025** | 0.00 |
| $\varepsilon = 5$ | **1.00*** | 0.40 | **0.375** | 0.075 |
| $\varepsilon = 10$ | **1.00*** | **1.00*** | **0.925*** | 0.025 |
| Non-private | **1.00** | **1.00** | **1.00** | **1.00** |

We also carry out experiments under independent Gaussian design. The details and more comprehensive discussion of the findings are deferred to Appendix D.2. In summary, in this case also, our algorithm enjoys greater utility under the strong signal case as shown in Table 2.

**Computational resources and license information :**   All the experiments were performed in the Great Lakes cluster with 16 cores and 10 GB RAM. ABESS package is distributed under GNU General Public License, Version 3. CVXPY package is distributed under Apache License, Version 2.

## 6   Conclusion

In this paper, we study the variable selection performance of BSS under the differential privacy constraint. In order to achieve (pure) differential privacy, we adopt the exponential mechanism and establish its high statistical utility guarantee in terms of exact model recovery. Furthermore, for computational efficiency, we design a MH random walk that provably mixes with the stationary distribution within a mixing time of the polynomial order in $(n, p, s)$. We also show that the samples from the MH random walk enjoy approximate DP while retaining a high utility guarantee with experimental underpinnings. In summary, as discussed in Section 1.1, we establish both high utility and efficient computational guarantee for our model selection algorithm under privacy constraints, which is in sharp contrast with the previous works in DP model selection literature. Moreover, the proposed MCMC method is generic enough to adopt in other models beyond linear structures. For example, one can use this technique under the setup of generalized linear models with likelihood loss as the utility function. Therefore, our method can be used in diverse domains including medical studies to fast-track scientific discoveries and promote the practice of responsible AI.

To this end, we also point out some of the open problems and future directions. One limitation, of our main result Theorem 3.5 is that it requires the condition $\min_{j \in \gamma^*} |\beta_j| = \Omega(\sqrt{(s \log p)/n})$ in high-privacy regime. It is still an open question whether the extra $\sqrt{s}$ factor is necessary for model selection. Future research along this line could focus on solving BSS through DP mixed integer optimization (MIO). This would mean an important contribution in this field as commercial solvers like GUROBI or MOSEK would be capable of solving the BSS problems at an industrial scale with high computational efficiency using a general DP framework.

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

# A  Proof of Main results

## A.1  Proof of Lemma 3.1

Recall that $u_K(\gamma; \mathbf{X}, \mathbf{y}) = -L_{\gamma,K}(\mathbf{X}, \mathbf{y})$ where $L_{\gamma,K}(\mathbf{X}, \mathbf{y}) := \sum_{i=1}^{n}(y_i - \mathbf{x}_{i,\gamma}^{\top}\boldsymbol{\beta})^2$. Therefore, it suffices to show that $L_{\gamma,K}(\cdot, \cdot)$ is data monotone. Let $D = \{(\mathbf{x}_i, y_i)\}_{i=1}^{n}$ and $D' = D \cup \{\mathbf{x}_{n+1}, y_{n+1}\}$. We define

$$\widehat{\boldsymbol{\beta}}_{n,\gamma} := \arg\min_{\boldsymbol{\beta}: \|\boldsymbol{\beta}\|_1 \leq K} \sum_{i=1}^{n}(y_i - \mathbf{x}_{i,\gamma}^{\top}\boldsymbol{\beta})^2,$$

$$\widehat{\boldsymbol{\beta}}_{n+1,\gamma} := \arg\min_{\boldsymbol{\beta}: \|\boldsymbol{\beta}\|_1 \leq K} \sum_{i=1}^{n+1}(y_i - \mathbf{x}_{i,\gamma}^{\top}\boldsymbol{\beta})^2.$$

Therefore, we have the following inequalities:

$$L_{\gamma,K}(D') = \sum_{i=1}^{n+1}(y_i - \mathbf{x}_{i,\gamma}^{\top}\widehat{\boldsymbol{\beta}}_{n+1,\gamma})^2 \geq \sum_{i=1}^{n}(y_i - \mathbf{x}_{i,\gamma}^{\top}\widehat{\boldsymbol{\beta}}_{n+1,\gamma})^2 \geq \sum_{i=1}^{n}(y_i - \mathbf{x}_{i,\gamma}^{\top}\widehat{\boldsymbol{\beta}}_{n,\gamma})^2 = L_{\gamma,K}(D).$$

The above inequalities conclude the proof.

## A.2  Proof of Sensitivity Bound (Lemma 3.3)

Let $(\mathbf{X}, \mathbf{y})$ and $(\tilde{\mathbf{X}}, \tilde{\mathbf{y}})$ be two neighboring datasets with $n$ and $n + 1$ observation respectively. For a subset $\gamma \in \mathscr{A}_s \cup \{\gamma^*\}$, consider the OLS estimators as follows:

$$\boldsymbol{\beta}_{\gamma,K} := \arg\min_{\boldsymbol{\theta}: \|\boldsymbol{\theta}\|_1 \leq K} \|\mathbf{y} - \mathbf{X}_{\gamma}\boldsymbol{\theta}\|_2^2, \quad \text{and} \quad \tilde{\boldsymbol{\beta}}_{\gamma,K} := \arg\min_{\boldsymbol{\theta}: \|\boldsymbol{\theta}\|_1 \leq K} \left\|\tilde{\mathbf{y}} - \tilde{\mathbf{X}}_{\gamma}\boldsymbol{\theta}\right\|_2^2.$$

From the definition of the score function $u(\gamma; \mathbf{X}, \mathbf{y})$, we have

$$u(\gamma; \mathbf{X}, \mathbf{y}) = -\sum_{i=1}^{n}(y_i - \mathbf{x}_{i,\gamma}^{\top}\boldsymbol{\beta}_{\gamma,K})^2$$

$$u(\gamma; \tilde{\mathbf{X}}, \tilde{\mathbf{y}}) = -\sum_{i=1}^{n}(y_i - \mathbf{x}_{i,\gamma}^{\top}\tilde{\boldsymbol{\beta}}_{\gamma,K})^2 - (\tilde{y}_{n+1} - \tilde{\mathbf{x}}_{n+1,\gamma}^{\top}\tilde{\boldsymbol{\beta}}_{\gamma,K})^2.$$

By the property of the OLS estimators, we have

$$u(\gamma; \mathbf{X}, \mathbf{y}) - u(\gamma; \tilde{\mathbf{X}}, \tilde{\mathbf{y}})$$
$$= \sum_{i=1}^{n}(y_i - \mathbf{x}_{i,\gamma}^{\top}\tilde{\boldsymbol{\beta}}_{\gamma,K})^2 + (\tilde{y}_{n+1} - \tilde{\mathbf{x}}_{n+1,\gamma}^{\top}\tilde{\boldsymbol{\beta}}_{\gamma,K})^2 - \sum_{i=1}^{n}(y_i - \mathbf{x}_{i,\gamma}^{\top}\boldsymbol{\beta}_{\gamma,K})^2$$
$$\leq \sum_{i=1}^{n}(y_i - \mathbf{x}_{i,\gamma}^{\top}\boldsymbol{\beta}_{\gamma,K})^2 + (\tilde{y}_{n+1} - \tilde{\mathbf{x}}_{n+1,\gamma}^{\top}\boldsymbol{\beta}_{\gamma,K})^2 - \sum_{i=1}^{n}(y_i - \mathbf{x}_{i,\gamma}^{\top}\boldsymbol{\beta}_{\gamma,K})^2$$
$$= (\tilde{y}_{n+1} - \tilde{\mathbf{x}}_{n+1,\gamma}^{\top}\boldsymbol{\beta}_{\gamma,K})^2$$
$$\leq (r + x_{\mathsf{max}}K)^2.$$

Similarly, we have $u(\gamma; \tilde{\mathbf{X}}, \tilde{\mathbf{y}}) - u(\gamma; \mathbf{X}, \mathbf{y}) \leq (r + x_{\mathsf{max}}K)^2$. Next, the $(\varepsilon, 0)$-DP follows from Lemma 2.2. This finishes the proof.

## A.3  Proof of Utility Guarantee (Theorem 3.5)

Consider the notation in Section B.1 and recall the event $\mathcal{E}_K := \cap_{\gamma: \gamma \in \mathscr{A}_s} \{\boldsymbol{\beta}_{\gamma,K} = \boldsymbol{\beta}_{\gamma,ols}\}$. We will For notational brevity, we use $L_\gamma$ to denote $L_\gamma(\mathbf{X}, \mathbf{y})$. Now, we restrict ourselves to the event $\mathcal{E}_K$. therefore we have $L_{\gamma,K} = L_\gamma$ for all $\gamma$. To establish a lower bound $\pi(\gamma^*)$, we make use of its specific form, thereby obtaining the following inequality:

$$\pi(\gamma^*) = \frac{1}{1 + \sum_{\gamma' \in \mathscr{A}_s} \exp\left\{-\frac{\varepsilon(L_{\gamma'} - L_{\gamma^*})}{\Delta u}\right\}}.$$

Now we fix $k \in [s]$, and and consider any $\gamma \in \mathscr{A}_{s,k}$. For any $\eta \in [0,1]$, note that

$$n^{-1}(L_\gamma - L_{\gamma^*}) = n^{-1}\{\mathbf{y}^\top(\mathbb{I}_n - \mathbf{\Phi}_\gamma)\mathbf{y} - \mathbf{y}^\top(\mathbb{I}_n - \mathbf{\Phi}_{\gamma^*})\mathbf{y}\}$$

$$= n^{-1}\left\{(\mathbf{X}_{\gamma^*\backslash\gamma}\boldsymbol{\beta}_{\gamma^*\backslash\gamma} + \mathbf{w})^\top(\mathbb{I}_n - \mathbf{\Phi}_\gamma)(\mathbf{X}_{\gamma^*\backslash\gamma}\boldsymbol{\beta}_{\gamma^*\backslash\gamma} + \mathbf{w}) - \mathbf{w}^\top(\mathbb{I}_n - \mathbf{\Phi}_{\gamma^*})\mathbf{w}\right\}$$

$$= \eta\boldsymbol{\beta}_{\gamma^*\backslash\gamma}^\top\Gamma(\gamma)\boldsymbol{\beta}_{\gamma^*\backslash\gamma} + 2^{-1}(1-\eta)\boldsymbol{\beta}_{\gamma^*\backslash\gamma}^\top\Gamma(\gamma)\boldsymbol{\beta}_{\gamma^*\backslash\gamma} - 2\left\{n^{-1}(\mathbb{I}_n - \mathbf{\Phi}_\gamma)\mathbf{X}_{\gamma^*\backslash\gamma}\boldsymbol{\beta}_{\gamma^*\backslash\gamma}\right\}^\top(-\mathbf{w})$$

$$+ 2^{-1}(1-\eta)\boldsymbol{\beta}_{\gamma^*\backslash\gamma}^\top\Gamma(\gamma)\boldsymbol{\beta}_{\gamma^*\backslash\gamma} - n^{-1}\mathbf{w}^\top(\mathbf{\Phi}_\gamma - \mathbf{\Phi}_{\gamma^*})\mathbf{w}.$$

Consider the random variable Following the analysis of Theorem 2.1 in [17], we have

$$\mathbb{P}\left[\max_{\gamma \in \mathscr{A}_{s,k}}\left|2n^{-1}\{(\mathbb{I}_n - \mathbf{\Phi}_\gamma)\mathbf{X}_{\gamma^*\backslash\gamma}\boldsymbol{\beta}_{\gamma^*\backslash\gamma}\}^\top\mathbf{w}\right| \geq 2^{-1}(1-\eta)\boldsymbol{\beta}_{\gamma^*\backslash\gamma}^\top\Gamma(\gamma)\boldsymbol{\beta}_{\gamma^*\backslash\gamma}\right] \leq 2e^{-6k\log p},$$

and,

$$\mathbb{P}\left[\max_{\gamma \in \mathscr{A}_{s,k}} n^{-1}\left|\mathbf{w}^\top(\mathbf{\Phi}_\gamma - \mathbf{\Phi}_{\gamma^*})\mathbf{w}\right| \geq 2^{-1}(1-\eta)\boldsymbol{\beta}_{\gamma^*\backslash\gamma}^\top\Gamma(\gamma)\boldsymbol{\beta}_{\gamma^*\backslash\gamma}\right] \leq 4e^{-2k\log p},$$

whenever

$$\frac{\min_{\gamma \in \mathscr{A}_{s,k}} \boldsymbol{\beta}_{\gamma^*\backslash\gamma}^\top\Gamma(\gamma)\boldsymbol{\beta}_{\gamma^*\backslash\gamma}}{k} \geq C\sigma^2\left\{\frac{\log p}{n(1-\eta)}\right\}$$

for large enough universal constant $C > 0$. Setting $\eta = 1/2$, we note that whenever $\mathfrak{m}_*(s) \geq 2C\sigma^2\{(\log p)/n\}$, we get

$$n^{-1}(L_\gamma - L_{\gamma^*}) \geq \frac{1}{2}\boldsymbol{\beta}_{\gamma^*\backslash\gamma}^\top\Gamma(\gamma)\boldsymbol{\beta}_{\gamma^*\backslash\gamma} \geq \frac{k\mathfrak{m}_*(s)}{2} \quad \text{for all } \gamma \in \mathscr{A}_s,$$

with probability at least $1 - 2p^{-6} - 4p^{-2}$. Also, note that $\boldsymbol{\beta}_{\gamma^*\backslash\gamma}^\top\Gamma(\gamma)\boldsymbol{\beta}_{\gamma^*\backslash\gamma} \leq \kappa_+ s b_{\max}^2$. Hence, if we have

$$\mathfrak{m}_*(s) \geq \max\left\{2C, \frac{16\Delta u}{\varepsilon\sigma^2}\right\}\frac{\sigma^2\log p}{n},$$

the following are true:

$$\sum_{\gamma' \in \mathscr{A}_s} \exp\left\{-\frac{\varepsilon(L_{\gamma'} - L_{\gamma^*})}{\Delta u}\right\}$$

$$\leq \sum_{\gamma' \in \mathscr{A}_s} \exp\left\{-\frac{nk\varepsilon\mathfrak{m}_*(s)}{2\Delta u}\right\}$$

$$\leq \sum_{k=1}^{s} \binom{p-s}{k}\binom{s}{k} \exp\left\{-\frac{nk\varepsilon\mathfrak{m}_*(s)}{2\Delta u}\right\}$$

$$\leq \sum_{k=1}^{s} p^{2k} \cdot p^{-4k} \leq p^{-2}.$$

Therefore, we have

$$\min_{\gamma \in \mathscr{A}_s \cup \{\gamma^*\}} \pi(\gamma) \geq \frac{1}{1+p^{-2}} \geq 1 - p^{-2}$$

with probability $1 - 2p^{-6} - 4p^{-2}$. Now by the discussion in Section B.1, we have $\mathbb{P}(\mathcal{E}_K) \geq 1 - 2p^{-2}$ for $K \geq \sqrt{s}\left\{\left(\frac{\kappa_+}{\kappa_-}\right)b_{\max} + \left(\frac{8}{\kappa_-}\right)\sigma x_{\max}\right\}$. This finishes the proof.

## A.4 Proof of Lemma 4.1

For clarity, we first specify some notations. Let $\gamma_t^D$ denote the model update of MH chain run over dataset $D$. Let $\tau_\eta^D$ be the corresponding $\eta$-mixing time. Let $D$ and $D'$ be two neighboring datasets, and $\pi^D$ and $\pi^{D'}$ be the corresponding probability mass functions for the exponential mechanism. Then, we have the following:

$$
\begin{aligned}
\mathbb{P}(\gamma_{\tau_\eta^D}^D = \gamma) &\leq \pi^D(\gamma) + \eta \\
&\leq e^\varepsilon \pi^{D'}(\gamma) + \eta \\
&\leq e^\varepsilon \mathbb{P}(\gamma_{\tau_\eta^{D'}}^{D'} = \gamma) + \eta(1 + e^\varepsilon).
\end{aligned}
$$

This finishes the proof.

## A.5 Proof of Mixing Time (Theorem 4.3)

We again restrict ourselves to the event $\mathcal{E}_K$ with $K \geq \sqrt{s}\left\{(\frac{\kappa_+}{\kappa_-})b_{\max} + (\frac{8}{\kappa_-})\sigma x_{\max}\right\}$. For the proof, let $\widetilde{\mathbf{P}}$ denote the transition matrix of the original Metropolis-Hastings sampler (10). In this case, the state space is $\mathscr{S} = \mathscr{A}_s \cup \{\gamma^*\}$. Now consider the transition matrix $\mathbf{P} = \widetilde{\mathbf{P}}/2 + \mathbb{I}_n/2$, corresponding to the lazy version of the random walk that stays in its current position with a probability of at least $1/2$. Due to the construction, the smallest eigenvalue of $\mathbf{P}$ is always non-negative, and the mixing time of the chain is completely determined by the second largest eigenvalue $\lambda_2$ of $\mathbf{P}$. To this end, we define the spectral gap $\mathsf{Gap}(\mathbf{P}) = 1 - \lambda_2$, and for any lazy Markov chain, we have the following sandwich relation [41, 50]

$$
\frac{1}{2}\frac{(1 - \mathsf{Gap}(\mathbf{P}))}{\mathsf{Gap}(\mathbf{P})}\log(1/(2\eta)) \leq \tau_\eta \leq \frac{\log[1/\min_{\gamma \in \mathscr{S}}\pi(\gamma)] + \log(1/\eta)}{\mathsf{Gap}(\mathbf{P})}. \tag{15}
$$

**Lower Bound on $\pi(\cdot)$ :**

To establish a lower bound on the target distribution in (7), we make use of its specific form, thereby obtaining the following inequality:

$$
\begin{aligned}
\pi(\gamma) &= \pi(\gamma^*).\frac{\pi(\gamma)}{\pi(\gamma^*)} \\
&= \frac{1}{1 + \sum_{\gamma' \in \mathscr{A}_s}\exp\left\{-\frac{\varepsilon(L_{\gamma'} - L_{\gamma^*})}{\Delta u}\right\}}\cdot\exp\left\{-\frac{\varepsilon(L_\gamma - L_{\gamma^*})}{\Delta u}\right\}.
\end{aligned}
$$

Now we fix $k \in [s]$, and and consider any $\gamma \in \mathscr{A}_{s,k}$.

Similar to the proof of Section A.3, we note that whenever $\mathfrak{m}_*(s) \geq 2C\sigma^2\{(\log p)/n\}$ for a large enough universal constant $C > 0$, we get

$$
\frac{3}{2}\boldsymbol{\beta}_{\gamma^*\backslash\gamma}^\top\Gamma(\gamma)\boldsymbol{\beta}_{\gamma^*\backslash\gamma} \geq n^{-1}(L_\gamma - L_{\gamma^*}) \geq \frac{1}{2}\boldsymbol{\beta}_{\gamma^*\backslash\gamma}^\top\Gamma(\gamma)\boldsymbol{\beta}_{\gamma^*\backslash\gamma} \geq \frac{k\mathfrak{m}_*(s)}{2} \quad \text{for all } \gamma \in \mathscr{A}_s,
$$

with probability at least $1 - 2p^{-6} - 4p^{-2}$. Also, note that $\boldsymbol{\beta}_{\gamma^*\backslash\gamma}^\top\Gamma(\gamma)\boldsymbol{\beta}_{\gamma^*\backslash\gamma} \leq \kappa_+ sb_{\max}^2$. Hence, if we have

$$
\mathfrak{m}_*(s) \geq \max\left\{2C, \frac{16\Delta u}{\varepsilon\sigma^2}\right\}\frac{\sigma^2\log p}{n},
$$

the following are true:

$$\sum_{\gamma' \in \mathscr{A}_s} \exp\left\{-\frac{\varepsilon(L_{\gamma'} - L_{\gamma^*})}{\Delta u}\right\}$$

$$\leq \sum_{\gamma' \in \mathscr{A}_s} \exp\left\{-\frac{nk\varepsilon \mathbb{m}_*(s)}{2\Delta u}\right\}$$

$$\leq \sum_{k=1}^{s} \binom{p-s}{k}\binom{s}{k} \exp\left\{-\frac{nk\varepsilon \mathbb{m}_*(s)}{2\Delta u}\right\}$$

$$\leq \sum_{k=1}^{s} p^{2k} \cdot p^{-4k} \leq p^{-2},$$

and,

$$\exp\left\{-\frac{\varepsilon(L_\gamma - L_{\gamma^*})}{\Delta u}\right\} \geq \exp\left\{-\frac{3n\varepsilon \boldsymbol{\beta}_{\gamma^* \setminus \gamma}^\top \Gamma(\gamma) \boldsymbol{\beta}_{\gamma^* \setminus \gamma}}{2\Delta u}\right\}$$

$$\geq \exp\left\{-\frac{3ns\varepsilon \kappa_+ b_{\mathsf{max}}^2}{2\Delta u}\right\}$$

Combining these two facts we have

$$\min_{\gamma \in \mathscr{A}_s \cup \{\gamma^*\}} \pi(\gamma) \geq \frac{1}{1+p^{-2}} \exp\left\{-\frac{3ns\varepsilon \kappa_+ b_{\mathsf{max}}^2}{2\Delta u}\right\} \geq \frac{1}{2} \exp\left\{-\frac{3ns\varepsilon \kappa_+ b_{\mathsf{max}}^2}{2\Delta u}\right\} \qquad (16)$$

with probability $1 - 2p^{-6} - 4p^{-2}$.

## Lower Bound on Spectral Gap:

Now it remains to prove a lower bound on the spectral gap $\mathsf{Gap}(\mathbf{P})$, and we do so via the canonical path argument [41]. We begin by describing the idea of a canonical path ensemble associated with a Markov chain. Given a Markov chain $\mathcal{C}$ with state space $\mathscr{S}$, consider the weighted directed graph $G(\mathcal{C}) = (V, E)$ with vertex set $V = \mathscr{S}$ and the edge set $E$ in which a ordered pair $e = (\gamma, \gamma')$ is included as an edge with weight $\mathbf{Q}(e) = \mathbf{Q}(\gamma, \gamma') = \pi(\gamma)\mathbf{P}(\gamma, \gamma')$ iff $\mathbf{P}(\gamma, \gamma') > 0$. A *canonical path ensemble* $\mathcal{T}$ corresponding to $\mathcal{C}$ is a collection of paths that contains, for each ordered pair $(\gamma, \gamma')$ of distinct vertices, a unique simple path $T_{\gamma, \gamma'}$ connecting $\gamma$ and $\gamma'$. We refer to any path in the ensemble $\mathcal{T}$ as a canonical path.

[41] shows that for any reversible Markov chain and nay choice of a canonical path ensemble $\mathcal{T}$, the spectral gap of $\mathbf{P}$ is lower bounded as

$$\mathsf{Gap}(\mathbf{P}) \geq \frac{1}{\rho(\mathcal{T})\ell(\mathcal{T})}, \qquad (17)$$

where $\ell(\mathcal{T})$ corresponds to the length of the longest path in the ensemble $\mathcal{T}$, and the quantity $\rho(\mathcal{T}) := \max_{e \in E} \frac{1}{Q(e)} \sum_{(\gamma, \gamma'): e \in T_{\gamma, \gamma'}} \pi(\gamma)\pi(\gamma')$ is known as the *path congestion parameter*.

Thus, it boils down to the construction of a suitable canonical path ensemble $\mathcal{T}$. Before going into further details, we introduce some working notations. For any two given paths $T_1$ and $T_2$:

- Their intersection $T_1 \cap T_2$ denotes the collection of overlapping edges.
- If $T_2 \subset T_1$, then $T_1 \setminus T_2$ denotes the path obtained by removing all the edges of $T_2$ from $T_1$.
- We use $\bar{T}_1$ to denote the reverse of $T_1$.
- If the endpoint of $T_1$ is same as the starting point of $T_2$, then $T_1 \cup T_2$ denotes the path obtained by joining $T_1$ and $T_2$ at that point.

We will now shift focus toward the construction of the canonical path ensemble. At a high level, our construction follows the same scheme as in [51].

**Canonical path ensemble construction:**

First, we need to construct the canonical path $T_{\gamma,\gamma^*}$ from any $\gamma \in \mathscr{S}$ to the true model $\gamma^*$. To this end, we introduce the concept of *memoryless* paths. We call a set $\mathcal{T}_M$ of canonical paths memoryless with respect to the central state $\gamma^*$ if

1. for any state $\gamma \in \mathscr{S}$ satisfying $\gamma \neq \gamma^*$, there exists a unique simple path $T_{\gamma,\gamma^*}$ in $\mathcal{T}_M$ connecting $\gamma$ and $\gamma^*$;

2. for any intermediate state $\tilde{\gamma} \in \mathscr{S}$ on any path $T_{\gamma,\gamma^*} \in \mathcal{T}_M$, the unique path connecting $\tilde{\gamma}$ and $\gamma^*$ is the sub-path of $T_{\gamma,\gamma^*}$ starting from $\tilde{\gamma}$ and ending at $\gamma^*$.

Intuitively, this memoryless property tells that for any intermediate step in any canonical path, the next step towards the central state does not depend on history. Specifically, the memoryless canonical path ensemble has the property that in order to specify the canonical path connecting any state $\gamma \in \mathscr{S}$ and the central state $\gamma^*$, we only need to specify the next state from $\gamma \in \mathscr{S} \setminus \{\gamma^*\}$, i.e., we need a transition function $\mathcal{G} : \mathscr{S} \setminus \{\gamma^*\} \to \mathscr{S}$ that maps the current state $\gamma$ to the next state. For simplicity, we define $\mathcal{G}(\gamma^*) = \gamma^*$ to make $\mathscr{S}$ as the domain of $\mathcal{G}$. For a more detailed discussion, we point the readers to Section 4 of [51]. We now state a useful lemma that is pivotal to the construction of the canonical path ensemble.

**Lemma A.1** ([51]). *If a function $\mathcal{G} : \mathscr{S} \setminus \{\gamma^*\} \to \mathscr{S}$ satisfies the condition $d_H(\mathcal{G}(\gamma), \gamma^*) < d_H(\gamma, \gamma^*)$ for any state $\gamma \in \mathscr{S} \setminus \{\gamma^*\}$, then $\mathcal{G}$ is a valid transition map.*

Using the above lemma, we will now construct the memoryless set of canonical paths from any state $\gamma \in \mathscr{S}$ to $\gamma^*$ by explicitly specifying a transition map $\mathcal{G}$. In particular, we consider the following transition function:

- If $\gamma \neq \gamma^*$, we define $\mathcal{G}(\gamma)$ to be $\gamma'$, whch is formed by replacing the least influential covariate in $\gamma$ with most influential covariate in $\gamma^* \setminus \gamma$. In notations, we have $\gamma_j' = \gamma_j$ for all $j \notin \{j_\gamma, k_\gamma\}$, $\gamma_{j_\gamma}' = 1$ and $\gamma_{k_\gamma}' = 0$, where $j_\gamma := \arg\max_{j \in \gamma^* \setminus \gamma} \left\| \mathbf{\Phi}_{\gamma \cup \{j\}} \mathbf{X}_{\gamma^*} \boldsymbol{\beta}_{\gamma^*} \right\|_2^2$ and $k_\gamma := \arg\min_{k \in \gamma \setminus \gamma^*} \left\| \mathbf{\Phi}_{\gamma \cup \{j\}} \mathbf{X}_{\gamma^*} \boldsymbol{\beta}_{\gamma^*} \right\|_2^2 - \left\| \mathbf{\Phi}_{\gamma \cup \{j\} \setminus \{k\}} \mathbf{X}_{\gamma^*} \boldsymbol{\beta}_{\gamma^*} \right\|_2^2$. Thus, the transition step involves a double flip which entails that $d_H(\mathcal{G}(\gamma), \gamma^*) = d_H(\gamma, \gamma^*) - 2$.

Due to Lemma A.1, it follows that the above transition map $\mathcal{G}$ is valid and gives rise to a unique memoryless set $\mathcal{T}_M$ of canonical paths connecting any $\gamma \in \mathscr{S}$ and $\gamma^*$.

Based on this, we are now ready to construct the canonical path ensemble $\mathcal{T}$. Specifically, due to memoryless property, two simple paths $T_{\gamma,\gamma^*}$ and $T_{\gamma',\gamma^*}$ share an identical subpath to $\gamma^*$ starting from their first common intermediate state. Let $T_{\gamma \cap \gamma'}$ denote the common sub-path $T_{\gamma \cap \gamma^*} \cap T_{\gamma' \cap \gamma*}$, and $T_{\gamma \setminus \gamma'} := T_{\gamma,\gamma^*} \setminus T_{\gamma \cap \gamma'}$ denotes the remaining path of $T_{\gamma,\gamma^*}$ after removing the segment $T_{\gamma \cap \gamma'}$. The path $T_{\gamma' \setminus \gamma}$ is defined in a similar way. Then it follows that $T_{\gamma \setminus \gamma'}$ and $T_{\gamma' \setminus \gamma}$ have the same endpoint. Therefore, it is allowed to consider the path $T_{\gamma \setminus \gamma'} \cup \bar{T}_{\gamma' \setminus \gamma}$.

We call $\gamma$ a *precedent* of $\gamma'$ if $\gamma'$ is on the canonical path $T_{\gamma;\gamma^*} \in \mathcal{T}$, and a pair of states $\gamma, \gamma'$ are *adjacent* if the canonical path $T_{\gamma,\gamma'}$ is $e_{\gamma,\gamma'}$, the edge connecting $\gamma$ and $\gamma'$. Next, for $\gamma \in \mathscr{S}$, define

$$\Lambda(\gamma) := \{\tilde{\gamma} \mid \gamma \in T_{\tilde{\gamma}, \gamma^*}\} \tag{18}$$

denote the set of all precedents. We denote by $|T|$ the length of the path $T$. The following lemma provides some important properties of the previously constructed canonical path ensemble.

**Lemma A.2.** *For any distinct pair $(\gamma, \gamma') \in \mathscr{S} \times \mathscr{S}$:*

*(a) We have*

$$|T_{\gamma,\gamma^*}| \leq d_H(\gamma, \gamma^*)/2 \leq s, \quad and$$

$$|T_{\gamma,\gamma'}| \leq \frac{1}{2}\{d_H(\gamma, \gamma^*) + d_H(\gamma', \gamma^*)\} \leq 2s.$$

*(b) If $\gamma$ and $\gamma'$ are adjacent and $\gamma$ is precedent of of $\gamma'$, then*

$$\{(\bar{\gamma}, \bar{\gamma}') \mid e_{\gamma,\gamma'} \in T_{\bar{\gamma},\bar{\gamma}'}\} \subset \Lambda(\gamma) \times \mathscr{S}.$$

*Proof.* For the first claim, let us first assume that $|T_{\gamma,\gamma^*}| = k$, i.e., $\mathcal{G}^k(\gamma) = \gamma^*$ for the appropriate transition map $\mathcal{G}$. Also, recall that $|\gamma| = |\gamma^*| = s$. Hence, due to an elementary iterative argument, it follows that

$$
\begin{aligned}
2s \geq d_H(\gamma, \gamma^*) = d_H(\mathcal{G}(\gamma), \gamma^*) + 2 \\
= d_H(\mathcal{G}^2(\gamma), \gamma^*) + 4 \\
\vdots \\
= 2k.
\end{aligned}
$$

Also, note that $|T_{\gamma,\gamma'}| \leq |T_{\gamma,\gamma^*}| + |T_{\gamma',\gamma^*}|$. Hence, the claim follows using the previous inequality.

For the second claim, note that for any pair $(\bar{\gamma}, \bar{\gamma}')$ such that $T_{\bar{\gamma},\bar{\gamma}'} \ni e_{\gamma,\gamma'}$, we have two possible options : (i) $e_{\gamma,\gamma'} \in T_{\bar{\gamma}\setminus\bar{\gamma}'}$, or (ii) $e_{\gamma,\gamma'} \in T_{\bar{\gamma}'\setminus\bar{\gamma}}$. As $\gamma$ is precedent of $\gamma'$, the only possibility that we have is $e_{\gamma,\gamma'} \in T_{\gamma\setminus\gamma'}$. This shows that $\gamma$ belongs to the path $T_{\bar{\gamma},\gamma^*}$ and $\bar{\gamma} \in \Lambda(\gamma)$. $\square$

According to Lemma A.2(b), the path congestion parameter $\rho(T)$ satisfies

$$
\rho(T) \leq \max_{(\gamma,\gamma')\in\Gamma_*} \frac{1}{\mathbf{Q}(\gamma,\gamma')} \sum_{\bar{\gamma}\in\Lambda(\gamma),\bar{\gamma}'\in\mathscr{S}} \pi(\bar{\gamma})\pi(\bar{\gamma}') = \max_{(\gamma,\gamma')\in\Gamma_*} \frac{\pi[\Lambda(\gamma)]}{\mathbf{Q}(\gamma,\gamma')}, \tag{19}
$$

where the set $\Gamma_* := \{(\gamma,\gamma') \in \mathscr{S} \times \mathscr{S} \mid T_{\gamma,\gamma'} = e_{\gamma,\gamma'}, \gamma \in \Lambda(\gamma')\}$. Here we used the fact that the weight function $\mathbf{Q}$ satisfies the reversibility condition $\mathbf{Q}(\gamma,\gamma') = \mathbf{Q}(\gamma',\gamma)$ in order to restrict the range of the maximum to pairs $(\gamma,\gamma')$ where $\gamma \in \Lambda(\gamma')$.

For the lazy form of the Metropolis-Hastings walk (10), we have

$$
\begin{aligned}
\mathbf{Q}(\gamma,\gamma') &= \pi(\gamma)\mathbf{P}(\gamma,\gamma') \\
&\geq \frac{1}{ps}\pi(\gamma)\min\left\{1, \frac{\pi(\gamma')}{\pi(\gamma)}\right\} \geq \frac{1}{ps}\min\left\{\pi(\gamma), \pi(\gamma')\right\}.
\end{aligned}
$$

Substituting this bound in (19), we get

$$
\begin{aligned}
\rho(T) &\leq ps \max_{(\gamma,\gamma')\in\Gamma_*} \frac{\pi(\Lambda(\gamma))}{\min\{\pi(\gamma), \pi(\gamma')\}} \\
&= ps \max_{(\gamma,\gamma')\in\Gamma_*} \left\{\max\left\{1, \frac{\pi(\gamma)}{\pi(\gamma')}\right\} \cdot \frac{\pi(\Lambda(\gamma))}{\pi(\gamma)}\right\}.
\end{aligned} \tag{20}
$$

In order to prove that $\rho(T) = O(ps)$ with high probability, it is sufficient to prove that the two terms inside the maximum are $O(1)$. To this end, we introduce two useful lemmas.

**Lemma A.3.** *Consider the event*

$$
\mathcal{A}_n = \left\{\max_{\gamma\in\mathscr{S},\ell\notin\gamma} \mathbf{w}^\top(\mathbf{\Phi}_{\gamma\cup\{\ell\}} - \mathbf{\Phi}_\gamma)\mathbf{w} \leq 12\sigma^2 s \log p\right\}
$$

*Then we have $\mathbb{P}(\mathcal{A}_n) \geq 1 - p^{-2}$.*

*Proof.* First note that $\mathbf{w}^\top(\mathbf{\Phi}_{\gamma\cup\{\ell\}} - \mathbf{\Phi}_\gamma)\mathbf{w} = (\mathbf{h}_{\gamma,\ell}^\top\mathbf{w})^2$ for an appropriate unit vector $\mathbf{h}_{\gamma,\ell}$ depending only upon $\mathbf{X}_\gamma$ and $\mathbf{X}_\ell$. By Sub-gaussian tail inequality, we have

$$
\mathbb{P}\left\{(\mathbf{h}_{\gamma,\ell}^\top\mathbf{w})^2 \geq t\right\} \leq 2e^{-\frac{t}{2\sigma^2}}.
$$

Setting $t = 12\sigma^2 s \log p$ and applying an union bound we get

$$
\begin{aligned}
\mathbb{P}\left\{\max_{\gamma\in\mathscr{S},\ell\notin\gamma}(\mathbf{h}_{\gamma,\ell}^\top\mathbf{w})^2 \geq 12\sigma^2 s \log p\right\} &\leq 2\binom{p}{s}(p-s)p^{-6s} \\
&\leq 2p^{-3s} \\
&\leq p^{-2}.
\end{aligned}
$$

$\square$

**Lemma A.4.** *Suppose that, in addition to the conditions in Theorem 4.3, the event $\mathcal{A}_n$ holds. Then for all $\gamma \neq \gamma^*$, we have*

$$\frac{\pi(\gamma)}{\pi(\mathcal{G}(\gamma))} \leq p^{-3}.$$

*Moreover, for all $\gamma$,*

$$\frac{\pi[\Lambda(\gamma)]}{\pi(\gamma)} \leq 2.$$

Therefore, both Lemma A.3 and Lemma A.4 give $\rho(T) \leq 2ps$ with probability $1 - p^{-2}$. Lemma A.2(a) suggests that $\ell(T) \leq 2s$. Therefore, Equation (17) shows that $\mathsf{Gap}(\mathbf{P}) \geq \frac{1}{4ps^2}$ with probability $1 - p^{-2}$. Finally, combining (16) and (15), we get the following with $1 - 8p^{-2}$

$$\tau_\eta \leq C_2 ps^2 \left( \frac{n\varepsilon\kappa_+ b_{\mathsf{max}}^2}{\left\{ r + (\frac{\kappa_+}{\kappa_-})b_{\mathsf{max}}x_{\mathsf{max}} + (\frac{\sigma}{\kappa_-})x_{\mathsf{max}}^2 \right\}^2} + \log(1/\eta) \right),$$

where $C_2 > 0$ is a universal constant. Finally, the proof is concluded by arguing that $\mathbb{P}(\mathcal{E}_K^c) \leq 2p^{-2}$.

# B  Proof of Auxiliary Results

## B.1  Constrained problem to unconstrained OLS problem

Now we are ready to bound $\|\boldsymbol{\beta}_{\gamma,K}\|_1$. Define the OLS estimator corresponding to the model $\gamma$ as

$$\boldsymbol{\beta}_{\gamma,ols} = \underbrace{(\frac{\mathbf{X}_\gamma^\top \mathbf{X}_\gamma}{n})^{-1} \frac{\mathbf{X}_\gamma^\top \mathbf{X}_{\gamma^*} \boldsymbol{\beta}_{\gamma^*}}{n}}_{:=\mathbf{u}_1} + \underbrace{(\frac{\mathbf{X}_\gamma^\top \mathbf{X}_\gamma}{n})^{-1} \frac{\mathbf{X}_\gamma^\top \mathbf{w}}{n}}_{:=\mathbf{u}_2}.$$

In this section, we will show that there exists a choice for $K$ such that the event $\mathcal{E}_K := \cap_{\gamma:|\gamma|=s}\{\boldsymbol{\beta}_{\gamma,ols} = \boldsymbol{\beta}_{\gamma,K}\}$ holds with high probability. By Holder's inequality we have $\|\mathbf{u}_1\|_2 \leq \left\|(\mathbf{X}_\gamma^\top \mathbf{X}_\gamma/n)^{-1}\right\|_{\mathsf{op}} \left\|\mathbf{X}_\gamma^\top \mathbf{X}_{\gamma^*}/n\right\|_{\mathsf{op}} \left\|\boldsymbol{\beta}_{\gamma^*}\right\|_2 \leq (\frac{\kappa_+}{\kappa_-})b_{\mathsf{max}}$. Hence, an application of Cauchy-Schwarz inequality directly yields that $\|\mathbf{u}_1\|_1 \leq 2(\frac{\kappa_+}{\kappa_-})\sqrt{s}b_{\mathsf{max}}$. Next, note that

$$\|\mathbf{u}_2\|_2 \leq \left\|(\frac{\mathbf{X}_\gamma^\top \mathbf{X}_\gamma}{n})^{-1}\right\|_2 \left\|\frac{\mathbf{X}_\gamma^\top \mathbf{w}}{n}\right\|_2 \leq \sqrt{s}\left\|(\frac{\mathbf{X}_\gamma^\top \mathbf{X}_\gamma}{n})^{-1}\right\|_2 \left\|\frac{\mathbf{X}_\gamma^\top \mathbf{w}}{n}\right\|_\infty \leq \sqrt{s}\left\|(\frac{\mathbf{X}_\gamma^\top \mathbf{X}_\gamma}{n})^{-1}\right\|_2 \left\|\frac{\mathbf{X}^\top \mathbf{w}}{n}\right\|_\infty.$$

Therefore, we get $\|\mathbf{u}_2\|_1 \leq \frac{s}{\kappa_-} \left\|\mathbf{X}^\top \mathbf{w}/n\right\|_\infty$. In order to upper bound the last term in the previous inequality, we define $D_{i,j} = \mathbf{X}[i,j]w_j$ for all $(i,j) \in [s] \times [n]$. Using the sub-Gaussian property of $w_j$, we have $\mathbb{E}(e^{\lambda w_j}) \leq e^{\lambda^2 x_{\mathsf{max}}^2 \sigma^2/2}$. Therefore, due to Hoeffding's inequality, we have

$$\mathbb{P}\left( \frac{1}{n}\big| \sum_{j \in [n]} D_{i,j} \big| \geq 8\sigma x_{\mathsf{max}} \sqrt{\frac{\log p}{n}} \right) \leq 2p^{-4}.$$

Note that $\left\|\mathbf{X}^\top \mathbf{w}/n\right\|_\infty = \max_{i \in [s]} n^{-1}\big| \sum_{j \in [n]} D_{i,j} \big|$. Hence, by simple union-bound argument, it follows that

$$\mathbb{P}\left( \max_{\gamma:|\gamma|=s} \left\|\frac{\mathbf{X}_\gamma^\top \mathbf{w}}{n}\right\|_\infty \geq 8\sigma x_{\mathsf{max}} \sqrt{\frac{\log p}{n}} \right) \leq 2p^{-4} \leq 2p^{-2}.$$

Thus, Assumption 3.4(c) yields that $\left\|\boldsymbol{\beta}_{\gamma,K}\right\|_1^2 \leq s \left\{ (\frac{\kappa_+}{\kappa_-})b_{\mathsf{max}} + (\frac{8}{\kappa_-})\sigma x_{\mathsf{max}} \right\}^2$. Therefore, if $K \geq \sqrt{s}\left\{ (\frac{\kappa_+}{\kappa_-})b_{\mathsf{max}} + (\frac{8}{\kappa_-})\sigma x_{\mathsf{max}} \right\}$ then $\mathbb{P}(\mathcal{E}_K) \geq 1 - 2p^{-2}$.

## B.2  Proof of Corollary 4.4

Based on Theorem 4.3, we have $\|\pi_t - \pi\|_{\mathsf{TV}} \leq eta$ with probability at least $1 - c_2 p^{-2}$ whenever $t$ is sufficiently large. Also, by Theorem 3.5, we know $\pi(\gamma^*) \geq 1 - p^{-2}$ with probability $1 - c_1 p^{-2}$. Therefore, we have $\pi_t(\gamma^*) \geq 1 - \eta - p^{-2}$ with probability at least $1 - (c_! + c_2)p^{-2}$. This finishes the proof.

# C   Proof of Lemma A.4

**part (a):**

Let $j_\gamma, k_\gamma$ be the indices defined in the construction of $\mathcal{G}(\gamma)$. The we have $\gamma' = \gamma \cup \{j_\gamma\} \setminus \{k_\gamma\}$. Let $\mathbf{v}_1 = (\boldsymbol{\Phi}_{\gamma \cup \{j_\gamma\}} - \boldsymbol{\Phi}_\gamma)\mathbf{X}_{\gamma^*}\boldsymbol{\beta}_{\gamma^*}$ and $\mathbf{v}_2 = (\boldsymbol{\Phi}_{\gamma \cup \{j_\gamma\}} - \boldsymbol{\Phi}_{\gamma'})\mathbf{X}_{\gamma^*}\boldsymbol{\beta}_{\gamma^*}$. Then Lemma C.1 guarantees that

$$\|\mathbf{v}_1\|_2^2 \geq n\kappa_- \mathfrak{m}_*(s), \quad \text{and} \quad \|\mathbf{v}_2\|_2^2 \leq n\kappa_- \mathfrak{m}_*(s)/2. \tag{21}$$

By the form in (7), we have

$$\frac{\pi(\gamma)}{\pi(\gamma')} = \exp\left\{-\frac{\mathbf{y}^\top(\boldsymbol{\Phi}_{\gamma'} - \boldsymbol{\Phi}_\gamma)\mathbf{y}}{(\Delta u/\varepsilon)}\right\}.$$

To show that the above ration is $O(1)$, it suffices to show that $\mathbf{y}^\top(\boldsymbol{\Phi}_{\gamma'} - \boldsymbol{\Phi}_\gamma)\mathbf{y}$ is large. By simple algebra, it follows that

$$\begin{aligned}
\mathbf{y}^\top(\boldsymbol{\Phi}_{\gamma'} - \boldsymbol{\Phi}_\gamma)\mathbf{y} &= \mathbf{y}^\top(\boldsymbol{\Phi}_{\gamma \cup \{j_\gamma\}} - \boldsymbol{\Phi}_\gamma)\mathbf{y} - \mathbf{y}^\top(\boldsymbol{\Phi}_{\gamma \cup \{j_\gamma\}} - \boldsymbol{\Phi}_{\gamma'})\mathbf{y} \\
&= \|\mathbf{v}_1\|_2^2 + 2\mathbf{v}_1^\top\mathbf{w} + \mathbf{w}^\top(\boldsymbol{\Phi}_{\gamma \cup \{j_\gamma\}} - \boldsymbol{\Phi}_\gamma)\mathbf{w} - \left\{\|\mathbf{v}_2\|_2^2 + 2\mathbf{v}_2^\top\mathbf{w} + \mathbf{w}^\top(\boldsymbol{\Phi}_{\gamma \cup \{j_\gamma\}} - \boldsymbol{\Phi}_{\gamma'})\mathbf{w}\right\} \\
&= \|\mathbf{v}_1\|_2^2 + 2\mathbf{v}_1^\top(\boldsymbol{\Phi}_{\gamma \cup \{j_\gamma\}} - \boldsymbol{\Phi}_\gamma)\mathbf{w} + \mathbf{w}^\top(\boldsymbol{\Phi}_{\gamma \cup \{j_\gamma\}} - \boldsymbol{\Phi}_\gamma)\mathbf{w} \\
&\quad - \left\{\|\mathbf{v}_2\|_2^2 + 2\mathbf{v}_2^\top(\boldsymbol{\Phi}_{\gamma \cup \{j_\gamma\}} - \boldsymbol{\Phi}_{\gamma'})\mathbf{w} + \mathbf{w}^\top(\boldsymbol{\Phi}_{\gamma \cup \{j_\gamma\}} - \boldsymbol{\Phi}_{\gamma'})\mathbf{w}\right\} \\
&\geq \|\mathbf{v}_1\|_2(\|\mathbf{v}_1\|_2 - 2\|(\boldsymbol{\Phi}_{\gamma \cup \{j_\gamma\}} - \boldsymbol{\Phi}_\gamma)\mathbf{w}\|_2) - \|\mathbf{v}_2\|_2(\|\mathbf{v}_2\|_2 + 2\|(\boldsymbol{\Phi}_{\gamma \cup \{j_\gamma\}} - \boldsymbol{\Phi}_{\gamma'})\mathbf{w}\|_2) \\
&\quad - \|(\boldsymbol{\Phi}_{\gamma \cup \{j_\gamma\}} - \boldsymbol{\Phi}_{\gamma'})\mathbf{w}\|_2^2.
\end{aligned} \tag{22}$$

Now, we recall the event

$$\mathcal{A}_n = \left\{\max_{\gamma \in \mathscr{S}, \ell \notin \gamma} \mathbf{w}^\top(\boldsymbol{\Phi}_{\gamma \cup \{\ell\}} - \boldsymbol{\Phi}_\gamma)\mathbf{w} \leq 12\sigma^2 s \log p\right\}.$$

Let $A^2 := n\kappa_- \mathfrak{m}_*(s) \geq \kappa_- C_0 \sigma^2 \log p$. Then for $C_0$ large enough so that $\kappa_- C_0 \geq (128 \times 12)s$, Equation (22) leads to the following inequality under event $\mathcal{A}_n$:

$$\mathbf{y}^\top(\boldsymbol{\Phi}_{\gamma'} - \boldsymbol{\Phi}_\gamma)\mathbf{y} \geq A(A - A/4) - (A/\sqrt{2})(A/\sqrt{2} + A/4) - A^2/16 \geq A/8.$$

This readily yields that

$$\frac{\pi(\gamma)}{\pi(\gamma')} \leq \exp\left\{-\frac{n\kappa_- \mathfrak{m}_*(s)}{(16\Delta u/\varepsilon)}\right\} \leq p^{-3} \tag{23}$$

under the margin condition of Theorem 4.3.

**Part (b):**

From the previous part, the bound (23) implies that $\pi(\gamma)/\pi(\mathcal{G}(\gamma)) \leq p^{-3}$. For each $\bar{\gamma} \in \Lambda(\gamma)$, we have that $\gamma \in T_{\bar{\gamma}, \gamma} \subset T_{\bar{\gamma}, \gamma^*}$. Let the path $T_{\bar{\gamma}, \gamma}$ be $\gamma_0 \to \gamma_1 \to \ldots \to \gamma_k$, where $k = |T_{\bar{\gamma}, \gamma}|$ is the length of the path, and $\gamma_0 = \bar{\gamma}$ and $\gamma_k = \gamma$ are the two endpoints. Now note that $\{\gamma_\ell\}_{\ell \leq k-1} \subset \mathscr{S}$, and (23) ensures that

$$\frac{\pi(\bar{\gamma})}{\pi(\gamma)} = \prod_{\ell=1}^k \frac{\pi(\gamma_{\ell-1})}{\pi(\gamma_\ell)} \leq p^{-3k}.$$

Also, by Lemma A.2(a) we have $k \in [s]$. Now, we count the total number of sets in $\Lambda(\gamma)$ for each $k \in [s]$. Recall that by the construction of the canonical path, we update the current state by adding a new influential covariate and deleting one unimportant one. Hence any state in $\mathscr{S}$ has at most $sp$ adjacent precedents, implying that there could be at most $s^k p^k$ distinct paths of length $k$. This entails that

$$\frac{\pi(\Lambda(\gamma))}{\pi(\gamma)} \leq \sum_{\bar{\gamma} \in \Lambda(\gamma)} \frac{\pi(\bar{\gamma})}{\pi(\gamma)} \leq \sum_{k=1}^s (ps)^k p^{-3k} \leq \sum_{k=1}^s p^{-k} \leq \frac{1}{1 - 1/p} \leq 2.$$

## C.1 Supporting lemmas

Recall the definition of $j_\gamma$ and $k_\gamma$. The first result in the following lemma shows that the gain in adding $j_\gamma$ to the current model $\gamma$ is at least $n\kappa_- \mathfrak{m}_*(s)$. The second result shows that the loss incurred by removing $k_\gamma$ from the model $\gamma \cup \{j_\gamma\}$ is at most $n\kappa_- \mathfrak{m}_*(s)/2$. As a result, it follows that it is favorable to replace $\mathbf{X}_{k_\gamma}$ with the more influential feature $\mathbf{X}_{j_\gamma}$ in the current model $\gamma$.

**Lemma C.1.** *Under Assumption 3.4(b) and Assumption 4.2, the following hold for all $\gamma \in \mathscr{A}_s$:*

*(a)* $\left\| \boldsymbol{\Phi}_{\gamma \cup \{j_\gamma\}} \mathbf{X}_{\gamma^*} \boldsymbol{\beta}_{\gamma^*} \right\|_2^2 - \left\| \boldsymbol{\Phi}_\gamma \mathbf{X}_{\gamma^*} \boldsymbol{\beta}_{\gamma^*} \right\|_2^2 \geq n\kappa_- \mathfrak{m}_*(s)$, *and*

*(b)* $\left\| \boldsymbol{\Phi}_{\gamma \cup \{j_\gamma\}} \mathbf{X}_{\gamma^*} \boldsymbol{\beta}_{\gamma^*} \right\|_2^2 - \left\| \boldsymbol{\Phi}_{\gamma \cup \{j_\gamma\} \setminus \{k\}} \mathbf{X}_{\gamma^*} \boldsymbol{\beta}_{\gamma^*} \right\|_2^2 \leq n\kappa_- \mathfrak{m}_*(s)/2.$

*Proof.* For each $\ell \in \gamma^* \setminus \gamma$, we have

$$
\begin{aligned}
\left\| \boldsymbol{\Phi}_{\gamma \cup \{\ell\}} \mathbf{X}_{\gamma^*} \boldsymbol{\beta}_{\gamma^*} \right\|_2^2 - \left\| \boldsymbol{\Phi}_\gamma \mathbf{X}_{\gamma^*} \boldsymbol{\beta}_{\gamma^*} \right\|_2^2 &= \boldsymbol{\beta}_{\gamma^*}^\top \mathbf{X}_{\gamma^*}^\top (\boldsymbol{\Phi}_{\gamma \cup \{\ell\}} - \boldsymbol{\Phi}_\gamma) \mathbf{X}_{\gamma^*} \boldsymbol{\beta}_{\gamma^*} \\
&= \frac{\boldsymbol{\beta}_{\gamma^*}^\top \mathbf{X}_{\gamma^*}^\top (\mathbb{I}_n - \boldsymbol{\Phi}_\gamma) \mathbf{X}_\ell \mathbf{X}_\ell^\top (\mathbb{I}_n - \boldsymbol{\Phi}_\gamma) \mathbf{X}_{\gamma^*} \boldsymbol{\beta}_{\gamma^*}}{\mathbf{X}_\ell^\top (\mathbb{I}_n - \boldsymbol{\Phi}_\gamma) \mathbf{X}_\ell} \\
&\geq \frac{\boldsymbol{\beta}_{\gamma^* \setminus \gamma}^\top \mathbf{X}_{\gamma^* \setminus \gamma}^\top (\mathbb{I}_n - \boldsymbol{\Phi}_\gamma) \mathbf{X}_\ell \mathbf{X}_\ell^\top (\mathbb{I}_n - \boldsymbol{\Phi}_\gamma) \mathbf{X}_{\gamma^* \setminus \gamma} \boldsymbol{\beta}_{\gamma^* \setminus \gamma}}{n},
\end{aligned}
$$

where the second equality simply follows from Gram-Schmidt orthogonal decomposition. By summing the preceding inequality over $\ell \in \gamma^* \setminus \gamma$, we get

$$
\begin{aligned}
\sum_{\ell \in \gamma^* \setminus \gamma} \left\| \boldsymbol{\Phi}_{\gamma \cup \{\ell\}} \mathbf{X}_{\gamma^*} \boldsymbol{\beta}_{\gamma^*} \right\|_2^2 - \left\| \boldsymbol{\Phi}_\gamma \mathbf{X}_{\gamma^*} \boldsymbol{\beta}_{\gamma^*} \right\|_2^2 &\geq \frac{\boldsymbol{\beta}_{\gamma^* \setminus \gamma}^\top \mathbf{X}_{\gamma^* \setminus \gamma}^\top (\mathbb{I}_n - \boldsymbol{\Phi}_\gamma) \mathbf{X}_{\gamma^* \setminus \gamma} \mathbf{X}_{\gamma^* \setminus \gamma}^\top (\mathbb{I}_n - \boldsymbol{\Phi}_\gamma) \mathbf{X}_{\gamma^* \setminus \gamma} \boldsymbol{\beta}_{\gamma^* \setminus \gamma}}{n} \\
&\geq \kappa_- \boldsymbol{\beta}_{\gamma^* \setminus \gamma}^\top \mathbf{X}_{\gamma^* \setminus \gamma}^\top (\mathbb{I}_n - \boldsymbol{\Phi}_\gamma) \mathbf{X}_{\gamma^* \setminus \gamma} \boldsymbol{\beta}_{\gamma^* \setminus \gamma} \\
&\geq n\kappa_- |\gamma \setminus \gamma^*| \mathfrak{m}_*(s) \\
&= n\kappa_- |\gamma^* \setminus \gamma| \mathfrak{m}_*(s).
\end{aligned}
$$

The last inequality follows from the fact that $|\gamma| = |\gamma^*| = s$. Since $j_\gamma$ maximizes $\left\| \boldsymbol{\Phi}_{\gamma \cup \{\ell\}} \mathbf{X}_{\gamma^*} \boldsymbol{\beta}_{\gamma^*} \right\|_2^2$ over all $\ell \in \gamma^* \setminus \gamma$, the preceding inequality implies that

$$
\left\| \boldsymbol{\Phi}_{\gamma \cup \{j_\gamma\}} \mathbf{X}_{\gamma^*} \boldsymbol{\beta}_{\gamma^*} \right\|_2^2 - \left\| \boldsymbol{\Phi}_\gamma \mathbf{X}_{\gamma^*} \boldsymbol{\beta}_{\gamma^*} \right\|_2^2 \geq n\kappa_- \mathfrak{m}_*(s).
$$

Similarly, to prove the second claim, first note that for any $k \in \gamma \setminus \gamma^*$, we have

$$
\begin{aligned}
\left\| \boldsymbol{\Phi}_{\gamma' \cup \{k\}} \mathbf{X}_{\gamma^*} \boldsymbol{\beta}_{\gamma^*} \right\|_2^2 - \left\| \boldsymbol{\Phi}_{\gamma'} \mathbf{X}_{\gamma^*} \boldsymbol{\beta}_{\gamma^*} \right\|_2^2 &= \boldsymbol{\beta}_{\gamma^* \setminus \gamma'}^\top \mathbf{X}_{\gamma^* \setminus \gamma'}^\top (\boldsymbol{\Phi}_{\gamma' \cup \{k\}} - \boldsymbol{\Phi}_{\gamma'}) \mathbf{X}_{\gamma^* \setminus \gamma'} \boldsymbol{\beta}_{\gamma^* \setminus \gamma'} \\
&= \frac{\boldsymbol{\beta}_{\gamma^* \setminus \gamma'}^\top \mathbf{X}_{\gamma^* \setminus \gamma'}^\top (\mathbb{I}_n - \boldsymbol{\Phi}_{\gamma'}) \mathbf{X}_k \mathbf{X}_k^\top (\mathbb{I}_n - \boldsymbol{\Phi}_{\gamma'}) \mathbf{X}_{\gamma^* \setminus \gamma'} \boldsymbol{\beta}_{\gamma^* \setminus \gamma'}}{\mathbf{X}_k^\top (\mathbb{I}_n - \boldsymbol{\Phi}_\gamma) \mathbf{X}_k} \\
&= \left\langle (\mathbb{I}_n - \boldsymbol{\Phi}_{\gamma'}) \mathbf{X}_{\gamma^* \setminus \gamma'} \boldsymbol{\beta}_{\gamma^* \setminus \gamma'}, \frac{(\mathbb{I}_n - \boldsymbol{\Phi}_{\gamma'}) \mathbf{X}_k}{\|(\mathbb{I}_n - \boldsymbol{\Phi}_{\gamma'}) \mathbf{X}_k\|_2} \right\rangle^2 \\
&\leq \left\| \boldsymbol{\beta}_{\gamma^* \setminus \gamma} \right\|_1^2 \left\| \frac{\mathbf{X}_{\gamma^* \setminus \gamma'}^\top (\mathbb{I}_n - \boldsymbol{\Phi}_{\gamma'}) \mathbf{X}_k}{\|(\mathbb{I}_n - \boldsymbol{\Phi}_{\gamma'}) \mathbf{X}_k\|_2} \right\|_\infty^2 \\
&\leq b_{\max}^2 \left\| \frac{\mathbf{X}_{\gamma^* \setminus \gamma'}^\top (\mathbb{I}_n - \boldsymbol{\Phi}_{\gamma'}) \mathbf{X}_k}{\|(\mathbb{I}_n - \boldsymbol{\Phi}_{\gamma'}) \mathbf{X}_k\|_2} \right\|_\infty^2.
\end{aligned}
$$

$\square$

Since $k_\gamma$ minimizes $\left\| \boldsymbol{\Phi}_{\gamma' \cup \{k\}} \mathbf{X}_{\gamma^*} \boldsymbol{\beta}_{\gamma^*} \right\|_2^2 - \left\| \boldsymbol{\Phi}_{\gamma'} \mathbf{X}_{\gamma^*} \boldsymbol{\beta}_{\gamma^*} \right\|_2^2$ over all possible $k \in \gamma \setminus \gamma^*$, by Assumption 4.2 we have

$$
\left\| \boldsymbol{\Phi}_{\gamma \cup \{j_\gamma\}} \mathbf{X}_{\gamma^*} \boldsymbol{\beta}_{\gamma^*} \right\|_2^2 - \left\| \boldsymbol{\Phi}_{\gamma \cup \{j_\gamma\} \setminus \{k\}} \mathbf{X}_{\gamma^*} \boldsymbol{\beta}_{\gamma^*} \right\|_2^2 \leq n\kappa_- \mathfrak{m}_*(s)/2.
$$

## D More simulation details

### D.1 Independent Uniform design

Under the setup of Section 5, we consider the privacy parameter $\varepsilon \in \{0.5, 1, 3, 5, 10\}$. For the Metropolis-Hastings random walk, we vary $K \in \{0.5, 2, 3, 3.5\}$ and initialize 10 independent Markov chains from random initializations and record the F-score of the last iteration. We also track the qualities of the model through its explanatory power. In particular, we calculate the scale factor $R_\gamma := \mathbf{y}^\top \mathbf{\Phi}_\gamma \mathbf{y} / \|\mathbf{y}\|_2^2$ for each model $\gamma \in \{\gamma_t\}_{t \geq 1}$ along the random walks. Typically, a high value of $R_\gamma$ will indicate the superior quality of the model $\gamma$. Note that $-\|\mathbf{y}\|_2^2 (1 - R_\gamma)$ is proportional to the log of the probability mass function function of $\gamma$. Thus, tracking $R_\gamma$ is equivalent to tracking the log-likelihood of $\gamma$ along the random walks.

**Strong signal:** Under this setup, note that the model estimate of ABESS exactly matches the true model. For $\varepsilon \geq 3$ and $K \geq 2$, Figure 1 shows that all the chains have identified a reasonably good estimate of the true model $\gamma^*$ within $50p$ iterations. This empirical phenomenon validates theoretical findings in Theorem 4.3. However, for larger values of $K$ the performance is worse as the noise level is also large. On the other hand, for the case of $K = 0.5$, the performance is also worse due to too much shrinkage that results in a bad estimate of $\boldsymbol{\beta}$. The mean F-score's also suggest the same phenomenon. For smaller values of $\varepsilon$, the performance is generally bad due to increased noise level. This is expected as higher privacy usually entails a worse performance in terms of utility.

**Weak signal:** We perform the same experiments under a weak signal regime. As expected, both Figure 2 and Table 1 show that the performance of the proposed algorithm is generally inferior to that in the strong signal regime for $K \geq 2$. However, note that our algorithm enjoys a better utility for $K = 0.5$. In fact, performance is as good as the non-private BSS for $\varepsilon \geq 3$. This is not surprising as $K = 0.5$ closer to $\|\boldsymbol{\beta}\|_1 \approx 0.7$ in the weak signal case and results in better estimation for $\boldsymbol{\beta}$. On the contrary, larger values of $K$ inject more noise into the algorithm and the utility deteriorates.

### D.2 Independent Gaussian Design

We consider an independent Gaussian design matrix, formed by sampling entries from identical independent standard normal distributions and normalized by the $\ell_\infty$ norm. Specifically, we set $n = 900, p = 2000$ with the sparsity level $s = 4$. Similar to the setup in Section 5, we generate entries with independent $\text{Uniform}(-0.1, 0.1)$ noise $\mathbf{w}$ following the linear model (1). We choose the design vector $\boldsymbol{\beta}$ with true sparsity $s = 4$ and the support set $\gamma^* = \{j : 1 \leq j \leq 4\}$. All the signal strengths are set to be equal, taking the following two forms: (i) **Strong signal:** $\beta_j = 2\{(s \log p)/n\}^{1/2}$, and (ii) **Weak signal:** $\beta_j = 2\{(\log p)/n\}^{1/2}$ for all $j \in \gamma^*$. We consider the privacy parameter $\varepsilon \in \{0.5, 1, 3, 5, 10\}$. For the Metropolis-Hastings random walk, we vary $K \in \{0.5, 2, 3, 3.5\}$ and initialize 10 independent Markov chains from random initialization and record the F-score of the last iteration.

**Strong signal:** Under this setup, note that the model estimate of ABESS exactly matches the true model with F-score $= 1$. For the case of $K = 0.5$, Figure 3 shows the performance is better compared with settings $K \geq 2$ when $\varepsilon \leq 5$. However, when $\varepsilon = 10$, the performance is worse due to shrinkage of the estimate of $\boldsymbol{\beta}$ while the estimations in other settings are easier due to lower privacy requirements. For higher values of $\varepsilon$, the performance is generally strong because of the reduced noise level. This is expected since lower privacy typically leads to better utility performance. Notice that for $\varepsilon = 10$ and $K = 2$, we have a fairly accurate estimate of the true model $\gamma^*$ within $50p$ iterations.

**Weak signal:** We conduct the same experiments under a weak signal regime. As expected, Figure 4 shows that the performance of the proposed algorithm is generally inferior to that in the strong signal regime for $K \geq 2$. However, note that our algorithm enjoys a better utility for $K = 0.5$ when $\varepsilon \geq 3$. In fact, performance is as good as the non-private BSS for $\varepsilon = 10$. This is not surprising as $K = 0.5$ closer to $\|\boldsymbol{\beta}\|_1 \approx 0.7$ in the weak signal case, leading to better estimation for $\boldsymbol{\beta}$. In contrast, larger values of $K$ introduce more noise into the algorithm and weaken the utility.

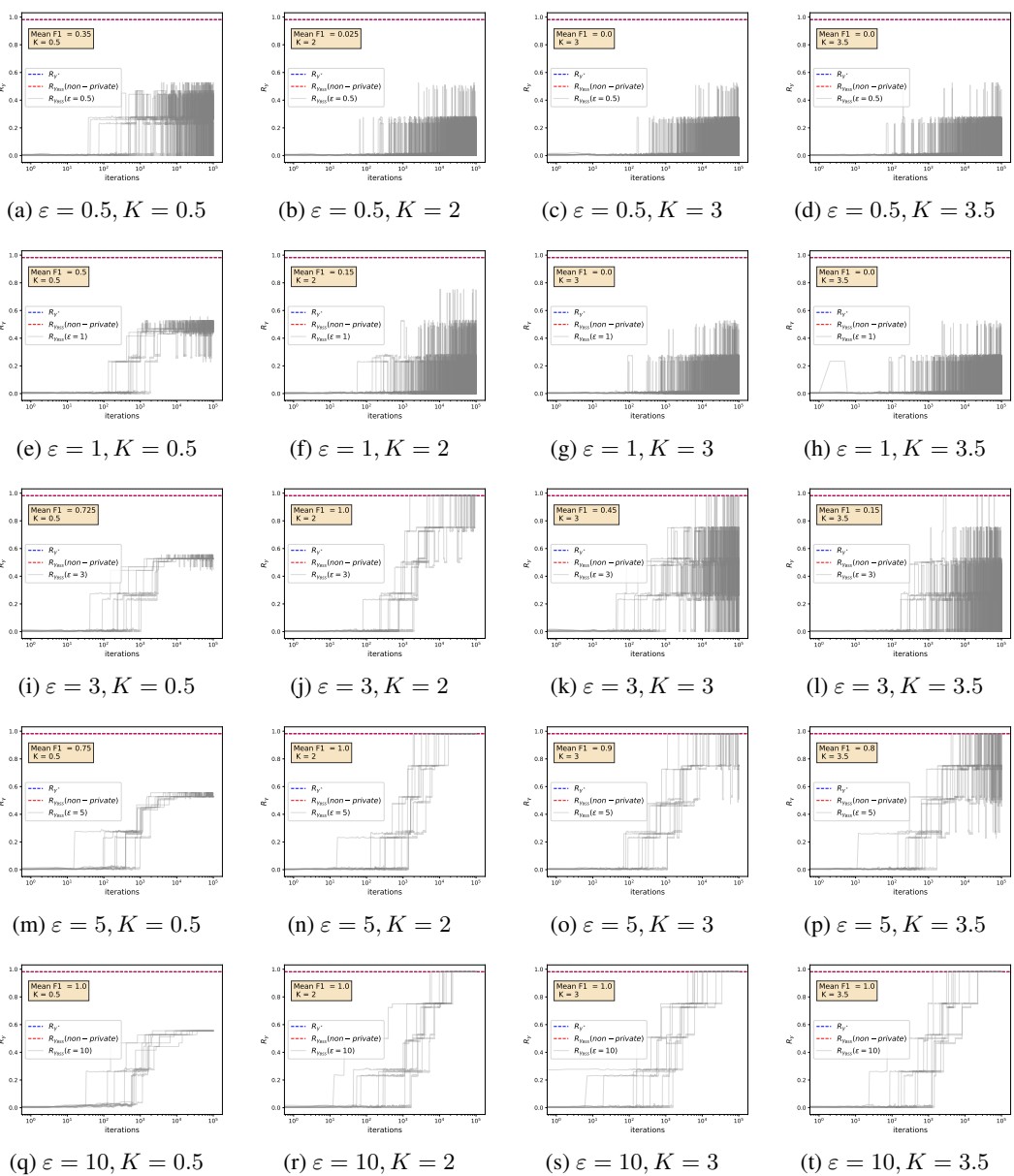

Figure 1: Metropolis-Hastings random walk under different privacy budgets and $\ell_1$ regularization. (Strong signal)

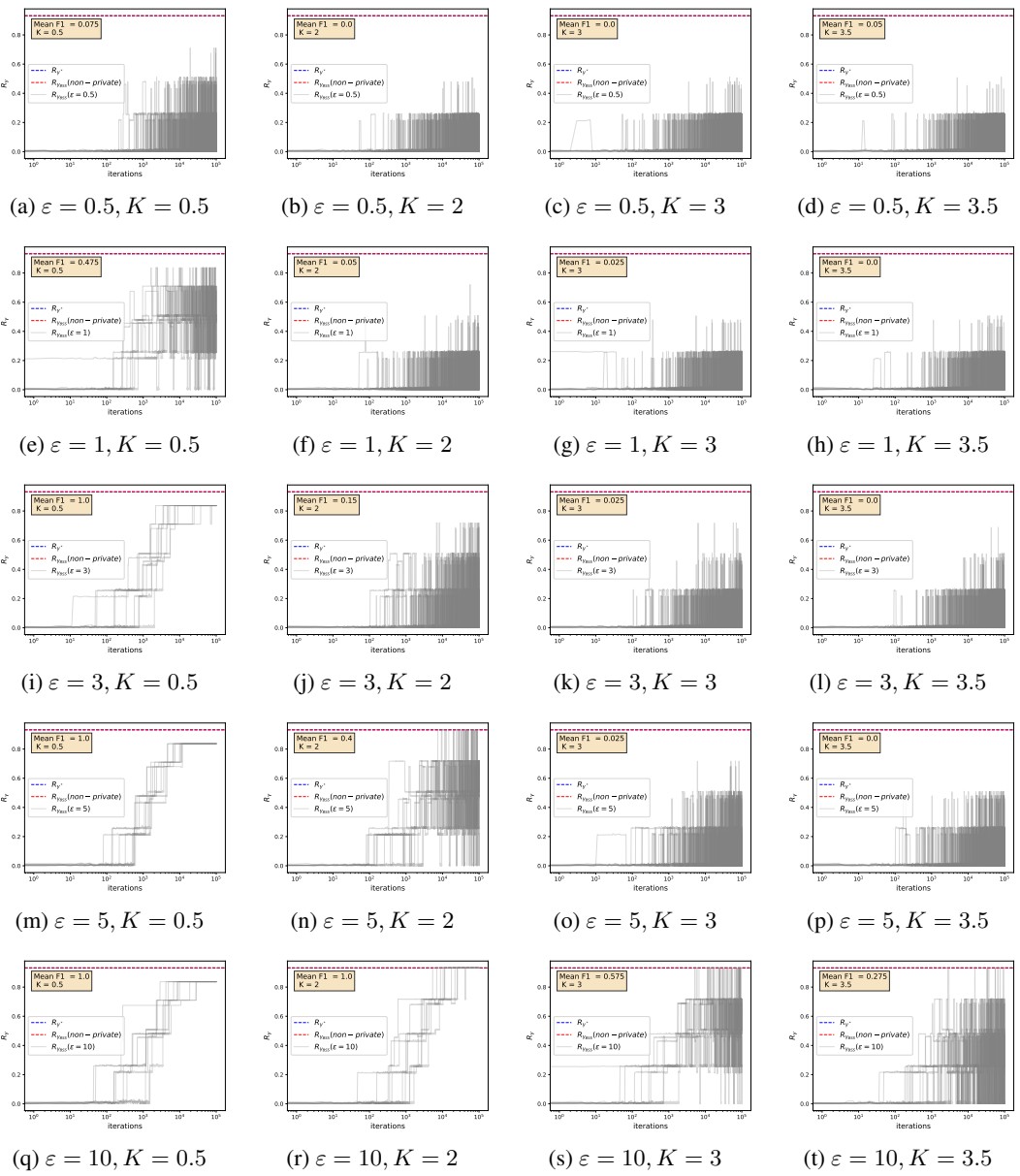

Figure 2: Metropolis-Hastings random walk under different privacy budgets and $\ell_1$ regularization. (Weak signal)

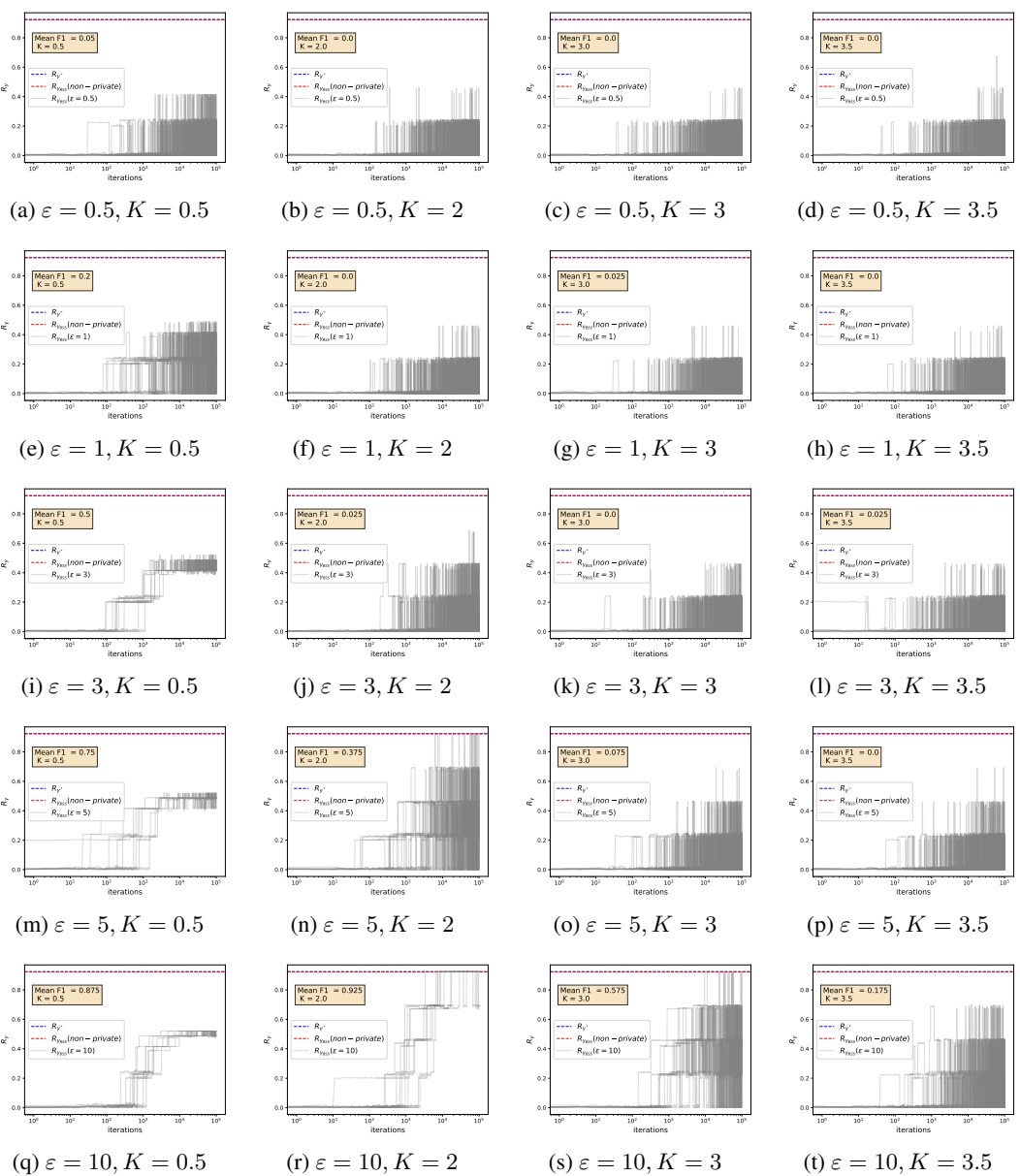

Figure 3: Gaussian setting Metropolis-Hastings random walk under different privacy budgets and $\ell_1$ regularization. (Strong signal)

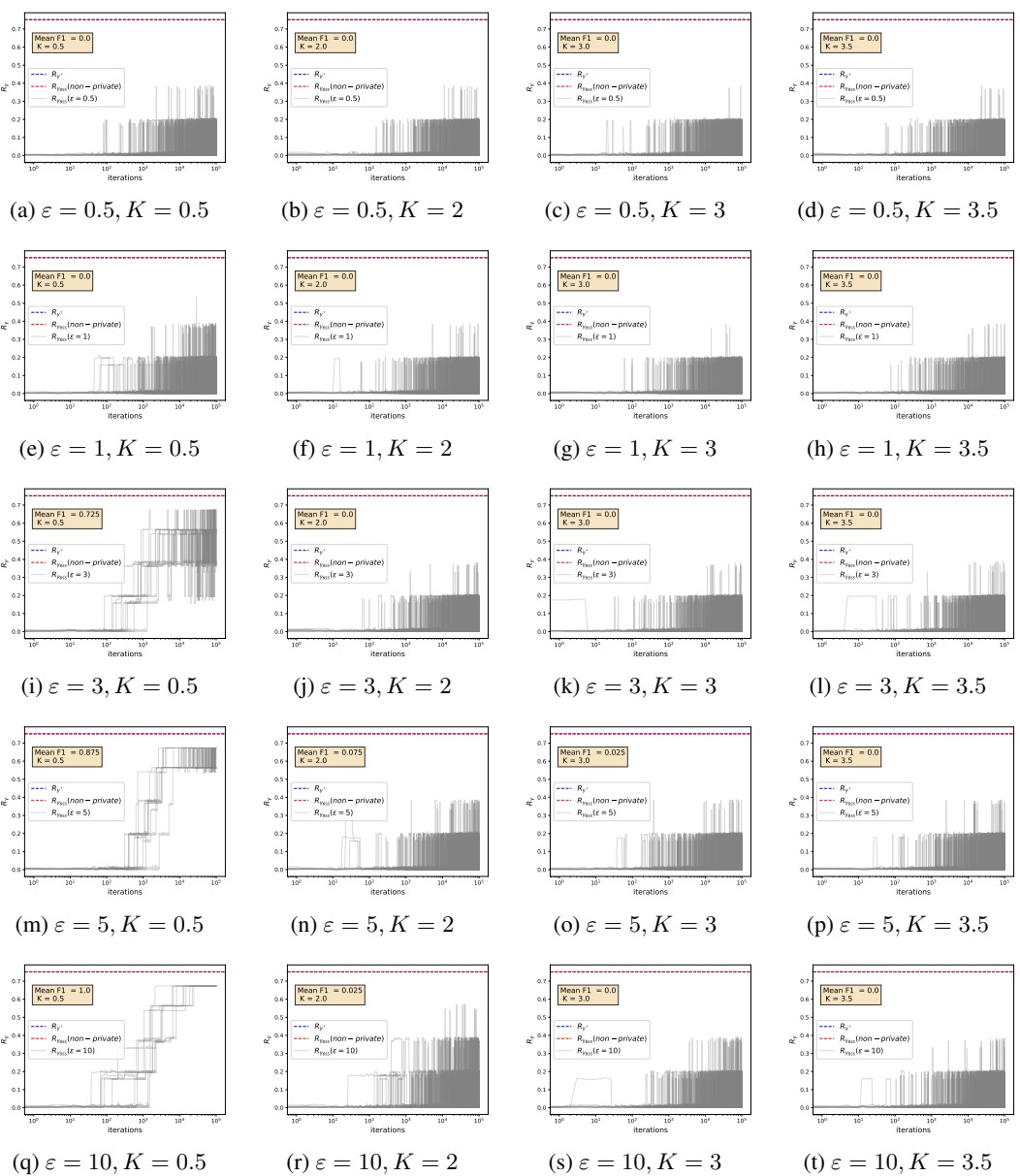

Figure 4: Gaussian setting Metropolis-Hastings random walk under different privacy budgets and $\ell_1$ regularization. (Weak signal)

