# OpenReview forum: "On the Computational Complexity of Private High-dimensional Model Selection"
_NeurIPS.cc/2024/Conference — NeurIPS 2024 poster_

### Official Review · Reviewer_piCf · 2024-07-08

**Soundness:** 3
**Presentation:** 3
**Contribution:** 3
**Rating:** 5
**Confidence:** 3

**Summary:**

This paper addresses the challenge of model selection in high-dimensional sparse linear regression models under privacy constraints. It proposes a differentially private algorithm for best subset selection using the exponential mechanism, providing strong statistical utility guarantees under high-privacy regimes. The authors demonstrate that accurate model recovery is possible in low-privacy regimes, achieving minimax optimal conditions similar to non-private settings. To tackle computational complexity, the paper introduces a Metropolis-Hastings algorithm with polynomial mixing time under certain conditions, ensuring approximate differential privacy and good statistical utility. Experiments on simulated data illustrate the algorithm's effectiveness in identifying active features under reasonable privacy budgets.

**Strengths:**

1. The proposed Exponential mechanism for the DP BSS problem improves $\beta_\min$ condition.
2. The proposed Metropolis-Hastings algorithm achieves approximate differential privacy with polynomial mixing time. The double swap update ensures that the proposed state maintains sparsity $s$.

**Weaknesses:**

1. The high-probability events in Theorem 3.5, Theorem 4.3, and Corollary 4.4 could be further clarified. See question 1.

**Questions:**

1. Theorem 3.5, Theorem 4.3, and Corollary 4.4 are high-probability statements. Is it possible to simplify them by including the high probability (success probability) $1 - c_i p^{-2}$ into the conclusion? For example, can the failure probability $1 - c_2 p^{-2}$ be included in the total variation distance to the stationary distribution in the mixing time?
2. In the proof of the utility guarantee (Theorem 3.5), the authors cite the analysis of Guo et al. (2020) titled "Best subset selection is robust against design dependence." Could the authors cite the specific theorems?
3. What is the disadvantage of output perturbation to address the DP BSS problem compared to the exponential mechanism (the authors propose)? Can output perturbation achieve pure DP with a good utility guarantee?

**Limitations:**

Please check the questions above.

---

> ### Author Rebuttal · Authors · 2024-08-01
>
> Thank you very much for recognizing the novelty and significance of the work. Below, we address all your concerns:
>
> **(P1)** To address Question 1, we first point out that both $\pi(\cdot)$ and $\pi_t(\cdot)$ are data-dependent distributions, i.e., these distributions depend on the data, and hence there is inherent randomness (due to noise $\boldsymbol{w}$) in them. Therefore, any statement on $\pi$ or $\pi_t$ can only be stated with a high success (or low failure) probability. One straightforward way to absorb the success/ failure probability in the conclusion is to consider the average performance. For example, Let $E:=\{\pi(\gamma^*) \ge 1- p^{-2}\}$. Then we have
>    $
>     \mathbb{E} [1-\pi(\gamma^*)] =\mathbb{E} [\{1 - \pi(\gamma^*)\} 1_E ] +  \mathbb{E}[ \{1 - \pi(\gamma^*)\} 1_{E^c}]= O (p^{-2}).
> $
>     Here we used the fact that $E^c$ has a small probability (Theorem 3.5).
>     This shows that $\mathbb{E}\{\pi(\gamma^*)\} \ge 1 - cp^{-2}$.
>     We can also present a similar argument to claim that $\mathbb{E} \{\mathrm{TV}(\pi_t, \pi)\} \le \eta + c_1 p^{-2}$ for large enough choices of $t$. However, we refrained from using these versions because the high-probability statements are easier to interpret.
>
> **(P2)** We would like to thank you for pointing out the lack of clarification in the proof of Theorem 3.5. We will add the particular theorem number of [1] in the revised version. Just to be clear, we were referring to Theorem 2.1 of [1].
>
> **(P3)** Recall that BSS outputs a model $\hat{S} \subseteq [p]$. One can also think of $\hat{S}$ as a binary vector $\hat{v}$ such that $\hat{v}_j = \mathbf{1}(j \in \hat{S})$. The main challenge lies here in the fact that $\hat{v}$ is discrete and the perturbed $\hat{v}$ should also remain in $\{0,1\}^p$. It seems very challenging to come up with a noise random variable $N$ that will ensure $\hat{v} + N \in \{0,1\}^p$ while maintaining $(\varepsilon, 0)$-DP. Moreover, it is crucial to study the stability of $\hat{S}$ for neighboring datasets to analyze the sensitivity $\hat{v}$, and this is challenging in general. Even if we figure out the sensitivity, we can not just use Laplace or Gaussian noise because it will destroy the sparsity structure in $\hat{v}$, and more importantly the perturbed vector will not be a binary vector anymore (i.e., loses its interpretability).
>
>   Another alternative way could be to just solve for $\hat{\beta}$ and then apply private top-k selection on $\hat{\beta}$. However, a good utility guarantee will require far worse assumptions like the minimal separation between the non-zero components of $\hat{\beta} = \Omega(s)$ (See Theorem 2.3 of [2] and the discussion afterward). This is definitely not true when all the non-zero components of $\hat{\beta}$ are of the order $ O(\sqrt{(\log p)/n})$. However, our method still achieves model consistency in such cases.
>
> [1] Yongyi Guo, Ziwei Zhu, and Jianqing Fan. Best subset selection is robust
> against design dependence. arXiv preprint arXiv:2007.01478, 2020
>
> [2] Gang Qiao, Weijie Su, and Li Zhang. Oneshot differentially private top-k se-
> lection. In International Conference on Machine Learning, pages 8672–8681.
> PMLR, 2021

---

> > ### Comment · Reviewer_piCf · 2024-08-14
> >
> > Thank you for your work on the rebuttal. I will keep my rating.

---

### Official Review · Reviewer_QHbD · 2024-07-12

**Soundness:** 3
**Presentation:** 3
**Contribution:** 2
**Rating:** 5
**Confidence:** 2

**Summary:**

The paper addresses the model selection problem in sparse linear regression, proposing a differentially private version of the best subset selection algorithm using the exponential mechanism. The authors prove that the proposed private algorithm requires $O(\sigma^2 s\log p/n\epsilon)$ samples to identify the correct features with high probability. Additionally, they demonstrate that when spurious features have low correlation with true features, sampling through MCMC converges rapidly to the stationary distribution.

**Strengths:**

- The paper is clearly written, and the proposed private algorithm is simple. The proposed method only requires common assumptions found in privacy and sparse linear regression literature. Based on these assumptions, the authors provide theoretical guarantees on both privacy and utility.
- The proposed private algorithm is of practical relevance as it can be implemented through MCMC. The authors also provide conditions under which MCMC achieves rapid convergence.

**Weaknesses:**

The rapid convergence of MCMC is achievable only when spurious features have low correlation with true features. However, practitioners typically do not know which features are true in advance, making it difficult to assess the strength of this correlation. Therefore, assumption 4.2 cannot be tested in practice.

**Questions:**

Does the universal constant $C_1$ in theorem 3.5 depend on the exponent of the failure probability?

**Limitations:**

See weaknesses.

---

> ### Author Rebuttal · Authors · 2024-08-01
>
> We greatly appreciate your encouraging words about our paper. Below we address your concerns point by point:
>
> **(P1)** We want to just briefly touch upon your comment regarding the weakness of the paper. You are absolutely right about the limitations of the correlation assumption (Assumption 4.2). Although the assumption needs the knowledge of the true support, similar assumptions are common in non-private high-dimensional model selection literature. For example, LASSO requires an irrepresentability condition which also controls the correlation between true and spurious features.
>
> **(P2)** To address your concern about $C_1$ and the exponent of failure probability, we would like to point you to Remark 3.6 of our paper. Essentially, to make the failure probability $O(p^{-M})$ for $M>2$, one needs to choose a larger constant $C_2 >C_1$.

---

### Official Review · Reviewer_2aDP · 2024-07-13

**Soundness:** 2
**Presentation:** 2
**Contribution:** 2
**Rating:** 4
**Confidence:** 2

**Summary:**

This papers study the differentially private best subset selection (BSS) of sparse linear model selection.

**Strengths:**

+ It presents an O(s), where s is the sparsity parameter, improvement in the utility-privacy tradeoff.

+ The MCMC implementation of exponential sampling seems a promising direction which may have some other independent applications. (I am not aware of the state-of-the-art results of exponential sampling and the MCMC study looks new to me and it seems that the authors claim it as a novel contribution of this paper).

**Weaknesses:**

- The comparisons with existing works are not comprehensive (see the comments below).

- The assumptions on both the sensitivity and margin need a better justification.

- The algorithm is based on computationally-intensive exponential sampling and it is not clear how practical the proposed MCMC improved version is.

- Code is not released and variation of the experimental results are marked.

**Questions:**

1. As partially mentioned by the authors in the introduction, there are many existing more computationally-efficient sparse optimizations. So compared to privately select a subset of coordinates in the first place, one may also apply L_1-norm restriction or applying sparse vector techniques as alternative solutions.  I am wondering whether the authors could give a more comprehensive studies on those existing works with a more clear motivation on the advantage of BSS.

2. Is it possible to remove the data-dependent assumption for bounded sensitivity (Assumption 3.2) but instead through clipping or truncation give an data-independent sensitivity bound?

3. I think the MCMC of exponential sampling could be an independent and possibly more important contribution of this paper, which may provide an efficient version of exponential sampling methods. I am wondering whether there is any prior work on this topic and more rigorously how to determine the constant in Theorem 4.3 and Corollary 4.4.

**Limitations:**

Please see the above comments.

---

> ### Author Rebuttal · Authors · 2024-08-01
>
> We thank you for your constructive comments. We will address all your concerns in the following points:
>
> **(P1)** ``***The assumptions on both the sensitivity and margin need a better justification***'' -- We did not make any assumption on the sensitivity. It is a consequence of Assumption 3.2. Please refer to point **(P3)** for a more detailed discussion.
>
>   We will add more discussion on the margin condition in the revised manuscript. The margin condition is rather necessary for BSS to achieve model consistency even in the non-private setting. Please refer to point **(P2)** in the response to reviewer QxEW for a more detailed discussion on this.
>
> **(P2)** ``***The algorithm is based on computationally intensive exponential sampling and it is not clear how practical the proposed MCMC improved version is.***''-- The exponential mechanism is a private algorithm and MCMC tries to mimic it in a computationally efficient way. Although the exponential mechanism has no advantage over the proposed MCMC, it is extremely important to understand its behavior to guarantee good performance of MCMC. The main proposed algorithm in the paper is the MCMC algorithm and to guarantee its good properties, a complete understanding of the exponential mechanism is required.
>
> **(P3)** ***Code is not released and variations of the experimental results are marked.*** - We did not release codes to maintain anonymity. We will definitely add a GitHub repository once the paper gets accepted.
>
> **(P4)** We would like to clarify that in this paper we specifically focus on subset selection rather than any generic sparse optimization.
>     You are right about the fact that we have mentioned a few prior existing subset selection algorithms that are computationally efficient. However, we have mentioned that they either demand very stringent $\beta_{min}$ conditions or have inferior utility guarantee. In contrast, our method achieves the best of three worlds: privacy, utility, and computation. [1] considered private $L_1$-constrained problem (LASSO) for model selection (discussed in Section 1.1), however, their utility guarantee is far worse compared to our algorithm. However, it is not entirely clear how one can use the sparse vector technique to solve subset selection problems in a sparse regression setting. One straightforward way to apply sparse vector technique (SVT) could be to first estimate  $\hat{\beta}$ using some non-private technique and then use SVT to select the active features.
>     However, this would potentially demand far stricter assumptions on design to guarantee control on $\Vert\hat{\beta} - \beta\Vert_\infty$. We can add a few relevant discussions in the revised version.
>
> **(P5)** The assumption on $y$ can be easily removed by considering truncation at level $R$. In this case, $R$ will be a tuning parameter of the algorithm and one needs to choose $R$ intelligently to achieve good utility.
>
> On the other hand, the boundedness assumption on $\boldsymbol{x}$ is very mild as one can always scale the design matrix.
>
> **(P6)** Thank you for recognizing the importance of the proposed MCMC algorithm. There has not been any prior work on exponential mechanisms in the model selection context that also studies the computation cost of the method. Also, the constants in Theorem 4.3 and Corollary 4.4 can be pinned down exactly by keeping track of the constants in the tail inequalities in the relevant proofs. We did not keep track of those because in high-dimensional literature it is more important to understand the effect of $n,p,s$.
>
>
> [1] Abhradeep Guha Thakurta and Adam Smith. Differentially private feature
> selection via stability arguments, and the robustness of the lasso. In Shai
> Shalev-Shwartz and Ingo Steinwart, editors, Proceedings of the 26th Annual
> Conference on Learning Theory, volume 30 of Proceedings of Machine Learning
> Research, pages 819–850, Princeton, NJ, USA, 12–14 Jun 2013. PMLR.

---

### Official Review · Reviewer_QExW · 2024-07-17

**Soundness:** 3
**Presentation:** 3
**Contribution:** 3
**Rating:** 5
**Confidence:** 3

**Summary:**

This paper studies the best subset selection (BSS) problem in high-dimensional sparse linear regression. The results of this paper are roughly as follows:
- First, adopt the exponential mechanism to design a DP BSS algorithm. The statistical/privacy guarantee of this approach can be derived based on standard techniques.
- Second, devise an efficient MCMC sampling algorithm that still enjoys approximate DP, while having a polynomial mixing time.
- Numerical experiments focus on a synthetic example, with a random design matrix, as well as a US census study and another socio-economic study.
Finally, the paper points out several open directions that would be plausible for future work.

**Strengths:**

- The paper provides a meticulous study of private BSS, including a detailed comparison of the signal strength needed to achieve the desired rates.
- The analysis involving rapid mixing of MCMC appears to be technically interesting, which could be a useful tool for future work.

**Weaknesses:**

- The simulation set up is somewhat restricted. Also, the main takeaways are not clear.

- The identifiablity margin assumption is not clearly explained; the intuition as to while this should be expected is not clear.

- In section 4, a comparison from the technical aspects is currently missing, and needs to be more clear to distinguish this work from prior works.

**Questions:**

- What is the main technical innovation of this work as compared to prior work?
- I understand that this paper is focused on BSS in high-dimensional linear regression. What are the practical takeaways of the results beyond this setup? Could you discuss it in this context?

**Limitations:**

The authors discuss limitations of their work in Section 6.

---

> ### Author Rebuttal · Authors · 2024-08-01
>
> We thank you for handling our paper and providing constructive comments. Below we provide point-by-point responses to your concerns:
>
> **(P1)** We will add more plots for the auto-regressive design to improve the experiment section. Current experiments show similar results as presented in the current paper. This establishes robustness in the performance of our proposed method. The main takeaway of the experiments is that large $\varepsilon$ generally improves the utility of our proposed method. However, the choice of $K$ also plays a key role in influencing the performance of the algorithm (discussed in Appendix A). Large $K$ usually entails that a large amount of noise is being injected into the algorithm, hence performance is generally poor for large $K$. Therefore, one needs to carefully choose $\varepsilon, K$ to get the best results. Also, under good choices of $\varepsilon$ and $K$, the plots do show that the chains mix very fast, which corroborates the results of Theorem 4.3.
>
> **(P2)** As discussed in Section 3.1, the margin $m_*(s)$ tries to capture both the degree of feature correlation and minimum signal strength simultaneously. If $m_*(s)$ is large, it essentially means that true features are easily distinguishable from the spurious features, and also the minimum signal strength is potentially very high. This is a general requirement for any method (including non-private methods) to achieve model consistency. Therefore, the condition Equation (9) is required to establish a model consistency result. The condition is not an assumption, but rather Theorem 3.5 says that if the condition holds then the exponential mechanism achieves model consistency. For example, under i.i.d. design, if the minimum signal strength is $\tilde{\Omega}(\sqrt{\log(p)/n})$ ($\tilde{\Omega}$ ignores some other terms) then Equation (9) is satisfied. See Remark 3.7 for a more detailed discussion.
>
> **(P3)** Thank you for your suggestion regarding adding comments on technical innovation. The main technical innovation in Section 4 of this paper is to design a **smart** MCMC chain that  **quickly** converges to the target distribution $\pi$. Moreover, the proposed algorithm achieves the best of three worlds: privacy, utility, and computation, which was not seen in prior works. We will add relevant discussions on this in Section 4 of the revised manuscript.
>
> **(P4)** The proposed method provides us with a framework for solving similar optimization problems involving discrete variables. For example, our method can be adopted in model selection problems in the generalized linear model (GLM). In this case, the objective function will just change, and essentially the same procedure can be used to privately solve the BSS problem in GLM setup. In these cases, the discrete variable is the support set $\hat{S}$. Therefore, our method can be used in a wide variety of other problems to obtain private solutions.

---

### Decision · Program_Chairs · 2024-09-25

**Decision:**

Accept (poster)

**Comment:**

The paper addresses the model selection problem in sparse linear regression, proposing a differentially private version of the best subset selection algorithm using the exponential mechanism. The authors show that their algorithm has a better privacy-utility (in terms of minimum value of coefficients required for identifiability) tradeoff than existing approaches. Additionally, they demonstrate that a simple MCMC algorithm for the problem converges efficiently under an additional condition (intuitively saying that spurious features have low correlation with true features). The main issue with the work is the required condition cannot be verified (without knowledge of true features) and privacy guarantees of the algorithm depend on it (since they depend on convergence). In addition, the empirical evaluation does not compare with any baselines. Overall, I recommend acceptance with some reservations.